# VAST: A Vision-Audio-Subtitle-Text Omni-Modality Foundation Model and Dataset

**Sihan Chen**[12]*, **Handong Li**[12]*, **Qunbo Wang**[2],
**Zijia Zhao**[21], **Mingzhen Sun**[21], **Xinxin Zhu**[2], **Jing Liu**[12]†

[1]School of Artificial Intelligence, University of Chinese Academy of Sciences
[2] Institute of Automation, Chinese Academy of Science
{sihan.chen, xinxin.zhu, jliu}@nlpr.ia.ac.cn,
{lihandong2023, qunbo.wang, zhaozijia2021, sunmingzhen2020}@ia.ac.cn

## Abstract

Vision and text have been fully explored in contemporary video-text foundational models, while other modalities such as audio and subtitles in videos have not received sufficient attention. In this paper, we resort to establish connections between multi-modality video tracks, including **V**ision, **A**udio, and **S**ubtitle, and **T**ext by exploring an automatically generated large-scale omni-modality video caption dataset called VAST-27M. Specifically, we first collect 27 million open-domain video clips and separately train a vision and an audio captioner to generate vision and audio captions. Then, we employ an off-the-shelf Large Language Model (LLM) to integrate the generated captions, together with subtitles and instructional prompts into omni-modality captions. Based on the proposed VAST-27M dataset, we train an omni-modality video-text foundational model named VAST, which can perceive and process vision, audio, and subtitle modalities from video, and better support various tasks including vision-text, audio-text, and multi-modal video-text tasks (retrieval, captioning and QA). Extensive experiments have been conducted to demonstrate the effectiveness of our proposed VAST-27M corpus and VAST foundation model. VAST achieves 22 new state-of-the-art results on various cross-modality benchmarks. Code, model and dataset will be released at https://github.com/TXH-mercury/VAST.

## 1 Introduction

A vast number of videos are recorded and uploaded to social media platforms every day, making the understanding of video content become a critical area of research in artificial intelligence. Simultaneously, the text serves as the most direct and efficient means of information propagation in our daily lives. Consequently, video-text cross-modality learning is crucial for AI systems to interact with humans, assisting them in understanding video content (video captioning), searching for desired videos (text-to-video retrieval), or answering questions about videos (video QA). With the advent of unsupervised pre-training techniques [1; 2] and large-scale video-text corpora [3; 4], video-text pre-training models have made rapid advancements and achieved relatively impressive performances on the aforementioned tasks. However, we contend that current pre-training models are far from perfect, as most of them are restricted to establishing connections solely between the text and visual content of videos, without incorporating other modality tracks such as audio and subtitles. Specifically, environmental audio can provide additional contexts overlapping with the visual information to reduce ambiguity and increase the prediction confidence of models, or

---

*Equal Contribution.
†Corresponding author.

37th Conference on Neural Information Processing Systems (NeurIPS 2023).

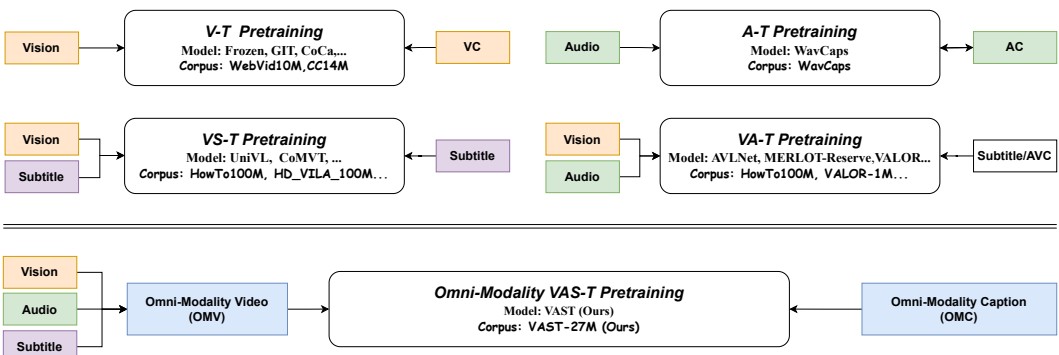

Figure 1: Illustration of the difference between conventional cross-modality pretraining and the proposed omni-modality pretraining. Thanks to the proposed omni-modality video caption corpus VAST-27M, the VAST foundation model can perceive videos from multiple information sources, including vision, audio, and subtitles, and enhance the connections between omni-modalities videos (OMV) and omni-modality captions (OMC) through large-scale pretraining. A, V, S, and T represent audio, vision, subtitle, and text, respectively. AC, VC, and AVC are abbreviations for audio, vision, and audiovisual captions.

complement the visual information to supply more multi-modal cues. Human speech or transcriptions (subtitles) also contain valuable information about conversations, news, or instructional procedures, which can be jointly modeled to enhance the comprehensive understanding of videos. Given that both audio and subtitles are informative for video comprehension, it is necessary to integrate them into a more advanced omni-modality foundational model, as depicted in Figure 1.

The primary challenge lies in the absence of an appropriate training corpus. To train such a foundational model, we require an omni-modality video caption corpus in which captions exhibit correspondence with vision, audio, and subtitles simultaneously. Unfortunately, as shown in Table 2, current video-text corpora either utilize raw subtitles as captions [4; 5; 6], contain only vision captions (alt-texts) [3], or possess audiovisual captions but limited in scale [7]. On the other hand, manual annotation of an omni-modality corpus is not feasible due to the exorbitant cost associated with the high demand for descriptions.

To address this issue, we propose a two-stage automatic pipeline to generate omni-modality captions for video clips from a large-scale open-domain video corpus (HD_VILA_100M [5]). Specifically, as depicted in Figure 2, in the first stage, we train separate vision and audio captioners on publicly available vision and audio caption corpora, which are then utilized to generate high-quality single-modality objective descriptions. In the second stage, we employ Vicuna-13b [8], an off-the-shelf Large Language Model (LLM), as a zero-shot omni-modality captioner. We feed the LLM with the generated single-modality captions, subtitles, and instructional prompts to encourage LLM to integrate different modality information and summarize them into a single long sentence, forming an omni-modality caption. Through this process, we create VAST-27M, a dataset that consists of 27 million video clips, each paired with 11 captions (including 5 vision, 5 audio, and 1 omni-modality caption). Given its comprehensive caption types, VAST-27M can contribute to various research communities, including vision-text, audio-text, and omni-modality pretraining. Extensive experiments have demonstrated that VAST-27M surpasses current video-text, audio-text, and audiovisual-text corpora by a significant margin.

Based on VAST-27M, we train a **V**ision-**A**udio-**S**ubtitle-**T**ext omni-modality foundational model named VAST. As shown in Figure 2, VAST consists of three single modality encoders while text encoder can fulfill cross-modality fusion through cross-attention layers. VAST is trained with three objectives including OM-VCC, OM-VCM, and OM-VCG (detailed in section 4.2) to enhance omni-modality understanding and generation capabilities. As shown in Table 1, compared to existing cross-modality pre-training models, VAST is trained on omni-modality video caption corpus and is capable of perceiving and processing all four modalities and supports a wide range of downstream tasks, with various modalities (vision-text, audio-text, and multi-modal video-text), and multiple types (retrieval, captioning, and QA).

Table 1: Comparisons between current cross-modality pretraining models and the proposed VAST. VAST is the first foundation model capable of perceiving four modalities and generalizing to various types of downstream tasks. V, A, S, and T represent vision, audio, subtitle, and text, respectively.

| Model | Task Type | | | Task Modality | | | | |
|---|---|---|---|---|---|---|---|---|
| | Retrieval | Discriminative | Generative | V-T | A-T | VA-T | VS-T | VAS-T |
| GIT [19] | | ✓ | ✓ | ✓ | | | | |
| VALUE [20] | ✓ | ✓ | ✓ | ✓ | | | ✓ | |
| VALOR [7] | ✓ | ✓ | ✓ | ✓ | ✓ | ✓ | | |
| VAST | ✓ | ✓ | ✓ | ✓ | ✓ | ✓ | ✓ | ✓ |

Our contributions are summarized as follows: (*i*) We introduce VAST-27M, the first large-scale omni-modality video caption dataset automatically generated by trained single modality captioners and Large Language Model; (*ii*) We train the first vision-audio-subtitle-text foundation model VAST which fulfills omni-modality perception and understanding; (*iii*) Extensive experiments verify the effectiveness of VAST and it outperforms state-of-the-art in a diverse range of cross-modality benchmarks.

## 2   Related Work

### 2.1   Cross-Modality Pretraining Corpus

**Video-Text Pretraining Corpus.** In the early stages, most video-text models were trained on the large-scale HowTo100M dataset [4]. This dataset consists of 136 million video clips from 1.22 million instructional YouTube videos, and employs automatically generated subtitles from ASR transcription tools as corresponding clip captions. Later, following similar schemes for collecting narrative videos-subtitles corpora, Zellers et al. proposed the YT-Temporal-180M corpus [6], which contains 180 million clips from 6 million YouTube videos. To overcome the constraint of the instructional domain, Xue et al. assembled the open-domain, high-resolution HD_VILA_100M corpus [5], consisting of 100 million clips from 3.3 million YouTube videos. However, while labeling subtitles is more cost-effective than annotating captions, subtitles often lack direct relevance to visual content, resulting in weak video-text correlations in model learning. Inspired by the collection schemes of the Conceptual Captions datasets (CC) [9; 10], Bain et al. compiled the WebVid2.5M and WebVid10M datasets [3], utilizing alt-texts as video captions, surpassing the aforementioned datasets that use subtitles as captions. However, alt-texts, although more related to video content than subtitles, differ in style from standard captions and still contain noise. Additionally, alt-texts typically describe visual content only, without incorporating other modalities. Taking a step further beyond vision, Chen et al. created the VALOR-1M dataset [7], which contains 1 million video clips from AudioSet [11] paired with annotated audiovisual captions. However, subtitle contents are not reflected in these captions, and the dataset's scale is limited and challenging to expand due to the expensive cost of manual annotation. In comparison to the aforementioned training corpora, our VAST-27M is the first high-quality, large-scale omni-modality video caption dataset whose captions encompass all modalities of video including vision, audio, and subtitles. In addition, its expansion is more cost-effective due to the automated generation pipeline.

**Audio-Text Pretraining Corpus.** In contrast to vision-text pretraining, audio-text pretraining has progressed relatively slowly due to limitations in training corpora, both in terms of quantity and scale. Human-labeled datasets such as Clotho [12], AudioCaps [13], MCAS [14], and AudioCaption [15] all contain fewer than 50,000 audio clips, which fall far short of the requirements for large-scale pretraining. Wu et al. compiled LAION-Audio-630K [16] by crawling audios and corresponding alt-texts from multiple sources, with a majority of audios sourced from Freesound [17]. Mei et al. introduced WavCaps [18], which comprises 403,000 audios and descriptions collected from various sources. They utilized ChatGPT to filter and rewrite the raw descriptions into caption-like sentences. However, the quality of the captions is heavily influenced by the quality of the raw descriptions. In contrast, we generates audio captions through a trained high-quality caption model, and VAST-27M is nearly two orders of magnitude larger than LAION-Audio-630K or WavCaps.

Table 2: Comparisons between current video-text and audio-text pretraining corpora and the proposed VAST-27M. VAST-27M is the first large-scale omni-modality video-caption corpus containing various types of captions generated automatically. S, VC, AC, AVC, and OMC stand for subtitle, vision, audio, audiovisual, and omni-modality captions, respectively.

| Dataset | Domain | #Clips | #Captions | Text Source | S | VC | AC | AVC | OMC |
|---|---|---|---|---|---|---|---|---|---|
| HowTo100M [4] | Instructional | 136M | 136M | ASR transcription | ✓ | | | | |
| YT_Temporal_180M [6] | Instructional | 180M | 180M | ASR transcription | ✓ | | | | |
| HD_VILA_100M [5] | Open | 103M | 103M | ASR transcription | ✓ | | | | |
| WebVid-2.5M [3] | Open | 2.5M | 2.5M | Alt-texts | | ✓ | | | |
| WebVid-10M [3] | Open | 10M | 10M | Alt-texts | | ✓ | | | |
| VALOR-1M [7] | Open | 1M | 1M | Manual | | | | ✓ | |
| LAION-Audio-630K [16] | Open | 634K | 634K | Alt-texts | | | ✓ | | |
| WavCaps [18] | Open | 403K | 403K | Generated | | | ✓ | | |
| VAST-27M (Ours) | Open | 27M | 297M | Generated | ✓ | ✓ | ✓ | | ✓ |

## 2.2 Multi-Modality Learning

There have been early methods exploring multi-modality tracks for video-text understanding. For instance, MMT [21] leverages seven experts to enhance text-to-video retrieval, while SMPFF [22] introduces audio for video captioning. In the context of video-text pretraining, there are works that jointly model subtitles or audio together with vision and text. Notable examples of incorporating subtitles include UniVL [23], CoMVT [24], and VALUE [20]. However, the subtitle-text correlations established in these models are implicit and weak, achieved through masked prediction [23] or next utterance prediction [24] that uses raw subtitles as prediction targets, rather than abstract text. This approach introduces inconsistency between the pretraining and fine-tuning stages. In contrast, VAST fully harnesses the generalization capabilities of Large Language Models to extract the most crucial information from subtitles into language descriptions. On the other hand, notable examples of incorporating audio include AVLNet [25], MERLOT Reserve [26], i-Code [27], and VALOR [7]. However, AVLNet, MERLOT Reserve, and i-Code focus on learning audio-subtitle relations rather than audio-text, limiting their generalization to tasks such as text-to-audio retrieval and audio captioning. While VALOR jointly models audio, vision, and text, it primarily targets perceiving environmental sounds and pays less attention to human speech. In comparison to the aforementioned methods, VAST is the first omni-modality foundation model capable of perceiving four modalities, connecting vision, audio, and subtitle to caption.

## 3 Dataset

### 3.1 Data Collection of VAST-27M

**Vision Captioner Training.** To establish the general correspondence between objects and visual concepts, we draw inspiration from BLIP [28] and commence by training a vision caption model on large-scale image-text corpora, including CC4M, CC12M, and a randomly selected set of 100M image-text pairs from LAION-400M [29]. Subsequently, we fine-tune the model on a combination of manually labeled image and video caption datasets, such as MSCOCO [30], VATEX [31], MSRVTT [32], and MSVD [33]. Through this process, we achieve a high-quality captioner capable of perceiving both static objects and dynamic actions, thus generating smooth and accurate captions.

**Audio Captioner Training.** For the audio captioning task, we train a dedicated audio caption model using a combination of large-scale audio-text corpora, namely VALOR-1M and WavCaps datasets. Unlike its vision counterpart, we do not employ a second-stage fine-tuning due to the limited scale of downstream audio caption datasets, which could lead to overfitting to a narrow range of audio concepts. Additionally, VALOR-1M is a human-annotated corpus that provides a certain level of assurance regarding the accuracy of the generated captions.

**Clip Selection.** Due to the consideration about time and money cost of downloading, storage and computation, we select a 27M video clips from the HD_VILA_100M dataset instead of using all of them. The selection rules are introduced as follows: 1) Clips whose length are shorter than 5 seconds or longer than 30 seconds are dropped. 2) Clips which have missing modalities of vision, audio or subtitle are dropped. 3) Clips are evenly chosen from the original 3.3M long-form videos in HD_VILA_100M.

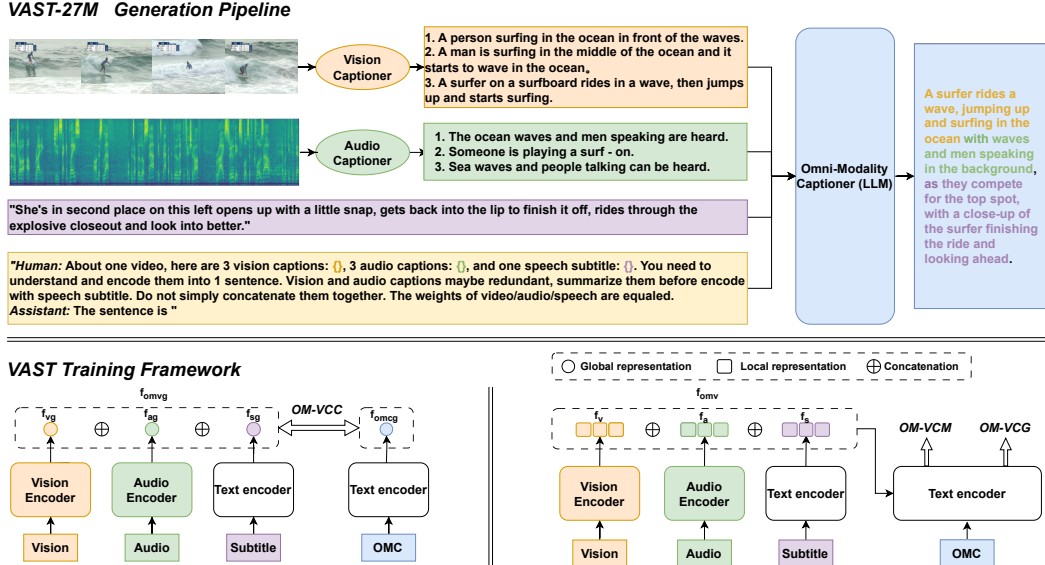

Figure 2: Illustration of the caption generation process of VAST-27M (top) and training framework of VAST (bottom). The vision and audio captioners generate captions based on the input video clip, and the Omni-Modality Captioner (Vicuna-13b) integrates them along with the raw subtitle and instructional prompts, to generate the omni-modality caption. The VAST model consists of three encoders and is trained under three objectives including OM-VCC, OM-VCM, and OM-VCG.

**Caption Generation.** For each video clip of VAST-27M, we employ the trained vision and audio captioners to generate 5 captions each, using a Top-K sampling approach with K=10. Subsequently, we utilize the off-the-shelf Vicuna-13b [8] model as the omni-modality captioner. Vicuna-13b is an open-source auto-regressive Large Language Model (LLM) based on the Transformer architecture, trained through fine-tuning of LLaMA [34] on user-shared conversations collected from ShareGPT. For each video clip, we randomly select 3 vision captions and 3 audio captions, and feed them, along with the raw subtitle and designed instructional prompts, into the LLM. As depicted in Figure 2, the LLM generates omni-modality captions that effectively describe the visual, audio, and subtitle contents, while adhering to a natural human captioning style. More examples of VAST-27M dataset can be found in Appendix.

## 3.2 Statistics of VAST-27M

VAST-27M consists of a total of 27M video clips sampled from the large-scale HD_VILA_100M corpus. The dataset covers 15+ categories, including music, gaming, education, entertainment, animals, and more, as opposed to being limited to instructional domains like HowTo100M. Compared to other commonly used cross-modality training corpora, as shown in Table 2, VAST-27M contains a larger number of captions (in terms of both modalities and quantity). The average lengths of vision, audio, and omni-modality captions in VAST-27M are 12.5, 7.2, and 32.4, respectively.

## 4 Approach

### 4.1 Basic Framework

As shown in Figure 2, VAST employs a fully end-to-end Transformer architecture, comprising a vision encoder (ViT [35]), an audio encoder (BEATs [36]), and a text encoder (BERT [2]). This framework can accommodate multi-modal inputs such as images, videos, audios, subtitles, and captions. The text encoder is responsible for encoding single-modality captions or subtitles, as well as performing multi-modal encoding/decoding through cross-attention layers. The omni-modality captions and subtitles are tokenized using the WordPiece tokenizer [37] and then fed into the text encoder to obtain the output features $f_{omc}$ and $f_s$, respectively. The vision encoder takes raw images

or sparsely sampled video frames as input and produces the output feature $f_v$. For audio clips, they are first divided into multiple 10-second-long clips, padded with zeros, converted to 64-dimensional log Mel filterbank spectrograms using a 25ms Hamming window, and then fed into the audio encoder to obtain the output feature $f_a$. The global representations ( [CLS] token feature) of these features are denoted as $f_{omcg}$, $f_{sg}$, $f_{vg}$, and $f_{ag}$, respectively.

## 4.2 Pretraining Objectives

Building upon conventional vision-text pretraining objectives, VAST employs the following three pretraining objectives:

***Omni-Modality Video-Caption Contrastive Loss (OM-VCC).*** The global omni-modality representations of video clips, denoted as $f_{omvg}$, are obtained by concatenating $f_{sg}$, $f_{vg}$, and $f_{ag}$. Subsequently, $f_{omvg}$ and $f_{omcg}$ are projected into the same semantic space using two linear layers and are then normalized. The contrastive loss is employed to regularize the feature distance between omni-modality video (OMV) and caption (OMC). The contrastive loss is defined as follows, where $sim(\cdot)$ represents the dot product of $f_{omcg}$ and $f_{omvg}$, and $B$ and $\tau$ denote the batch size and a learnable parameter, respectively.

$$\mathcal{L}_{\text{OM-VCC}} = -\frac{1}{2} \sum_{i=1}^{B} \log \frac{\exp(\tau \cdot sim(v_i, c_i))}{\sum_{j=1}^{B} \exp(\tau \cdot sim(v_i, c_j)))} - \frac{1}{2} \sum_{i=1}^{B} \log \frac{\exp(\tau \cdot sim(v_i, c_i)))}{\sum_{j=1}^{B} \exp(\tau \cdot sim(v_j, c_i)))} \tag{1}$$

***Omni-Modality Video-Caption Matching Loss (OM-VCM).*** This loss encourages the model to infer whether a pair of OMV and OMC is matched or not. Specifically, caption tokens are fed into the text encoder again, this time with cross-attention layers activated to attend to the condition features $f_{omv}$ which is obtained by concatenating those unpooled features $f_s$, $f_v$, and $f_a$ along the sequential dimension. Before concatenation, three independent linear layers are applied to adjust their hidden size to the same value. The output feature of the text encoder is then fed into a two-layer MLP to make binary predictions. To create informative negative pairs, we adopt a hard negative mining strategy following [38]. The loss function is formulated as follows, where $y = 1$ if OMV and OMC are matched, and 0 otherwise.

$$\mathcal{L}_{\text{OM-VCM}} = \mathbb{E}_{(v_i, c_i) \sim (\mathcal{V}, \mathcal{C})} \left[ y \log p_{vcm} + (1 - y) \log (1 - p_{vcm}) \right] \tag{2}$$

***Omni-Modality Video Caption Generation Loss (OM-VCG).*** This loss employs conditional causal masked language modeling to enhance the model's ability to generate omni-modality captions. Specifically, 60% of the tokens in OMC are masked at the input of the text encoder. Cross-attention layers are activated and $f_{omv}$ is used as condition features as in OM-VCM. The self-attention layers in the text encoder utilize a single-directional causal attention mask to prevent information leakage, and the masked tokens are reconstructed at the output of the text encoder using BERT's vanilla prediction layer. The loss is defined as follows, where $c_m$ and $c_{<m}$ denote the masked tokens and the tokens before them, respectively.

$$\mathcal{L}_{\text{OM-VCG}} = -\mathbb{E}_{(v_i, c_i) \sim (\mathcal{V}, \mathcal{C})} \log P \left( c_m \mid c_{<m}, v \right) \tag{3}$$

The overall loss $\mathcal{L}_{\text{OM}}$ is the sum of the three losses, with equal weights assigned for simplicity.

$$\mathcal{L}_{\text{OM}} = \mathcal{L}_{\text{OM-VCC}} + \mathcal{L}_{\text{OM-VCM}} + \mathcal{L}_{\text{OM-VCG}} \tag{4}$$

## 4.3 Modality Grouping

While VAST establishes omni-modality video-caption correspondence during pretraining, it is important to address the potential absence of modalities in downstream benchmarks and practical applications, due to that inconsistencies between the modalities used in pretraining and adaptation can have negative effects. Inspired by the modality grouping strategy proposed by VALOR [7], we uniformly model the relations of V-T, A-T, VA-T, VS-T, and VAS-T (previously introduced as $\mathcal{L}_{\text{OM}}$). Specifically, vision and audio captions are used in V-T and A-T modeling, respectively, while omni-modality captions are employed in VA-T, VS-T, and VAS-T modeling. The final loss is formulated as follows:

$$\mathcal{L} = \mathcal{L}_{\text{OM}} + \mathcal{L}_{\text{V-T}} + \mathcal{L}_{\text{A-T}} + \mathcal{L}_{\text{VA-T}} + \mathcal{L}_{\text{VS-T}} \tag{5}$$

Table 3: Performance comparison between VAST and state-of-the-art methods. VAST has achieved **22 new SOTA results**. Recall@1, CIDEr, and Acc are used as evaluation metrics for retrieval, captioning, and QA tasks, respectively. For captioning tasks, results marked with '*' means that SCST finetuning [41] is employed. 'MM benchmark' means that either audio or subtitle tracks are available in those benchmarks, and SOTA_Vis and SOTA_MM are the best methods with vision track used only or employing multi-modal tracks. **More detailed benchmark introductions, model references and comparison results can be found in Appendix**.

| Vision-Text benchmark | T-to-V Retrieval | | | Captioning | QA | | |
|---|---|---|---|---|---|---|---|
| | COCO[30] | Flickr[42] | Flickr(ZS)[42] | COCO[30] | TGIF[43] | MSVD[44] | VQAv2[45] |
| SOTA | **68.3**[46] | 90.3[47] | 89.7[46] | **154.9**\*[48] | 78.7[7] | 60.2[49] | **84.3**[50] |
| VAST | 68.0 | **91.0**(+0.7) | **90.4**(+0.7) | 149.0* | **79.1**(+0.4) | **60.2** | 80.2 |

| Audio-Text benchmark | T-to-A Retrieval | | | Captioning | | |
|---|---|---|---|---|---|---|
| | Clothov1[12] | Clothov2[12] | AudioCaps[13] | Clothov1[12] | Clothov2[12] | AudioCaps[13] |
| SOTA | 17.5[7] | 21.5[18] | 42.2[18] | 42.3[7] | 48.8[18] | **78.7**[18] |
| VAST | **25.1**(+7.6) | **26.9**(+5.4) | **52.0**(+9.8) | **50.7**(+8.4) | **51.9**(+3.1) | 78.2 |

| MM Video-Text benchmark | T-to-V Retrieval | | | | | |
|---|---|---|---|---|---|---|
| | MSRVTT[32] | YouCook2[51] | VALOR-32K[7] | VATEX[31] | DiDeMo[52] | ANET[53] |
| SOTA_Vis | 58.8[54] | 33.7[55] | 43.4[56] | 71.1[57] | 70.4[54] | 66.8[54] |
| SOTA_MM | 54.4[7] | 31.3[20] | 73.2[7] | 76.9[7] | 57.6[7] | 63.4[7] |
| VAST | **63.9**(+5.1) | **50.4**(+16.7) | **80.0**(+6.8) | **83.0**(+6.1) | **72.0**(+1.6) | **70.5**(+3.7) |

| MM Video-Text benchmark | Captioning | | | | | QA | | |
|---|---|---|---|---|---|---|---|---|
| | MSRVTT[32] | YouCook2[51] | VALOR-32K[7] | VATEX[31] | TVC[58] | MSRVTT[44] | MUSIC[59] | ANET[60] |
| SOTA_Vis | 75.9[19] | 131.2[19] | 27.3[61] | 94.5*[19] | 66.1[19] | 49.5[49] | - | 47.9[54] |
| SOTA_MM | 74.0[7] | 190.0[55] | 61.5[7] | 95.8*[7] | 66.0[62] | 49.2[7] | 78.9[7] | 48.6[7] |
| VAST | **78.0**(+2.1) | **198.8**(+8.8) | **62.0**(+0.5) | **99.5**\*(+3.7) | **74.1**(+8.0) | **50.1**(+0.6) | **80.7**(+1.8) | **50.4**(+1.8) |

# 5 Experiments

## 5.1 Implementation Details

VAST is trained using the PyTorch framework on 64 Tesla V100 cards. The vision, audio, and text encoders are initialized from EVAClip-ViT-G [39], BEATs, and BERT-B, respectively, resulting in a total parameter size of 1.3B. The training is conducted on a combination corpus consisting of VAST-27M, VALOR-1M, WavCaps, CC14M, and 110M randomly sampled pairs from LAION-400M, for a total of 200K training steps. At each training step, one corpus is sampled for training. In addition, the raw captions of CC14M and LAION are replaced with captions generated by the trained vision captioner. The initial learning rate is set to 1e-4, and a linear decay schedule is used. The batch size is set to 1024. During the pertaining stage, one video frame and two 10s long audio clips are randomly sampled for each video clip. For ablation studies, CLIP-ViT-B [40] is used as the vision encoder, with its parameters frozen to increase efficiency. Further details such as the mixture ratio of the pretraining dataset and the downstream finetuning configurations can be found in the Appendix.

Regarding the adaptation to downstream tasks, for retrieval tasks, all candidates are ranked using VCC, and then the Top-50 candidates are reranked using VCM. For captioning tasks, beam search with a beam size of 3 is employed. For QA tasks, they are formulated as open-ended generative problems, where questions serve as prefixes, and answers are predicted without any constraints. For SOTA comparison and ablation studies in the main paper, Recall@1, CIDEr, and Acc are used as evaluation metrics for retrieval, captioning, and QA tasks, respectively.

## 5.2 Comparison to State-of-the-Art Models

We present a comparison of the VAST foundation model with state-of-the-art models across various vision-text, audio-text, and multi-modal video-text benchmarks. The corresponding results are summarized in Table 3. Although the primary focus of VAST lies in enhancing omni-modality understanding and generation capabilities, it also demonstrates remarkable performance in image-text,

Table 4: Comparisons of V-T quality in VAST-27M and open-sourced vision-text corpora.

| Model | PT Data | MSVD | | | MSRVTT | | |
|---|---|---|---|---|---|---|---|
| | | RET | CAP | QA | RET | CAP | QA |
| (a) | - | 23.7 | 101.2 | 46.8 | 35.4 | 58.9 | 43.6 |
| (b) | WebVid2.5M | 43.1 | 139.2 | 53.0 | 42.4 | 63.7 | 45.5 |
| (c) | CC14M | 46.1 | 139.2 | 53.9 | 45.2 | 66.4 | 46.3 |
| (d) | LAION-150M | 42.4 | 139.3 | 53.2 | 44.9 | 66.9 | 45.4 |
| (e) | VALOR-1M | 42.3 | 136.7 | 52.6 | 42.8 | 65.3 | 45.5 |
| (f) | VAST-27M (subtitle) | 37.3 | 124.5 | 40.3 | 40.3 | 61.9 | 45.1 |
| (g) | VAST-27M (omc) | 46.8 | 143.2 | 54.3 | 47.5 | 67.4 | 46.7 |
| (h) | VAST-27M (vision caption) | **47.7** | **149.6** | **55.3** | **49.1** | **68.9** | **46.8** |

Table 5: Comparisons of A-T quality between VAST-27M and open-sourced audio-text corpora.

| Model | PT Data | Clothov2 | | AudioCaps | |
|---|---|---|---|---|---|
| | | RET | CAP | RET | CAP |
| (a) | - | 17.2 | 42.1 | 41.1 | 70.4 |
| (b) | VALOR-1M | 21.0 | 47.3 | 47.4 | 74.8 |
| (c) | WavCaps | 22.0 | 47.2 | 43.2 | 73.0 |
| (d) | VAST-27M (audio caption) | **23.1** | **48.9** | **47.4** | **76.9** |

vision-only video-text, and audio-text benchmarks. Specifically, VAST surpasses BEiT-3 [47] and BLIP-2 [46] on the text-to-image retrieval benchmark of Flickr30K under finetuning and zero-shot settings, respectively, establishing new state-of-the-art results in TGIF-QA and MSVD-QA benchmarks. Furthermore, VAST exhibits exceptional performance in audio-text benchmarks, achieving five new state-of-the-art results with significant improvements, due to the abundant generated captions in the VAST-27M corpus.

In the context of multi-modal video-text benchmarks, they can be classified into three categories: subtitle-oriented (YouCook2, TVC) benchmarks that require a comprehensive understanding of subtitles for reasoning, audio-oriented (VALOR-32K) benchmarks that necessitates careful attention to audio cues, and even-oriented benchmarks (others) where both audio and subtitles provide supportive roles to vision. Our results demonstrate that VAST excels across all types of benchmarks, outperforming previous both vision-only and multi-modal state-of-the-art models. Notably, we surpass the GIT2 [19] foundation model, specialized in captioning tasks, by 2.1, 67.6, 5.0, and 8.0 CIDEr points on MSRVTT, YouCook2, VATEX, and TVC captioning benchmarks, respectively, while utilizing only 22.5% of its parameters and 3.4% of its training data size. Furthermore, we achieve significant margins over the vision-audio-text foundation model VALOR [7] on 11 benchmarks. More detailed comparison results can be found in Appendix.

### 5.3 Comparison to Open-Source Cross-Modality Training Corpus

In this subsection, we quantitatively compare the quality of the proposed VAST-27M to current open-source video, audio, and audiovisual pretraining corpora. We achieve this by training models on these corpora and fine-tuning them on different types of downstream tasks, including video retrieval (RET), captioning (CAP), and question answering (QA).

**V-T Quality.** We train multiple V-T models on different corpora and fine-tune them on the MSVD and MSRVTT datasets. It is important to note that only the visual contents in videos are utilized for both pretraining and fine-tuning, and all models are trained for 50K steps. As shown in Table 4, the model trained with vision captions in VAST-27M (h) achieves the best results on all six benchmarks, outperforming the baselines and models trained with other corpora by a significant margin. In contrast, model (f), which treats subtitles as captions and can be viewed as raw HD_VILA_100M dataset, performs the worst due to the weak vision-text relations. Model trained on VAST-27M (g) outperforms other corpora but is still inferior to model (h), as the multi-modal captions introduce noise when only vision contents are fed into the model.

Table 6: Comparisons of OMV-OMC quality between VAST-27M and VALOR-1M.

| Model | PT Data | MSRVTT | | | YouCook2 | | VALOR-32K | |
|---|---|---|---|---|---|---|---|---|
| | | RET | CAP | QA | RET | CAP | RET | CAP |
| (a) | - | 37.6 | 60.3 | 44.6 | 32.6 | 120.9 | 37.8 | 40.8 |
| (b) | VALOR-1M | 45.4 | 67.8 | 46.4 | 29.3 | 120.1 | **70.8** | **53.9** |
| (c) | VAST-27M (omc w/o LLM) | 44.4 | 71.4 | 47.7 | 36.9 | 159.0 | 49.9 | 48.1 |
| (d) | VAST-27M (omc) | **53.0** | **71.7** | **47.9** | **44.0** | **187.5** | 58.8 | 48.5 |

Table 7: Downstream performances of models trained on VAST-27M with different modalities used during training and finetuning.

| Model | PT | FT | MSRVTT | | | YouCook2 | | VALOR-32K | |
|---|---|---|---|---|---|---|---|---|---|
| | | | RET | CAP | QA | RET | CAP | RET | CAP |
| (a) | - | V | 35.4 | 58.7 | 43.7 | 6.2 | 77.0 | 29.1 | 36.6 |
| (b) | - | V+A | 37.3(+1.9) | 61.1(+2.4) | 44.6(+0.9) | 7.1(+0.9) | 74.1(-2.9) | 38.4(+9.3) | 40.2(+3.6) |
| (c) | - | V+S | 35.8(+0.4) | 59.1(+0.4) | 43.6(-0.1) | 31.2(+25.0) | 119.1(+42.1) | 29.6(+0.5) | 38.1(+1.5) |
| (d) | - | V+A+S | 37.6(+2.2) | 60.3(+1.6) | 44.6(+0.9) | 32.6(+26.4) | 120.9(+43.9) | 37.8(+8.7) | 40.8(+4.2) |
| (e) | V | V | 47.5 | 67.4 | 46.7 | 14.0 | 93.6 | 56.5 | 45.0 |
| (f) | V | V+A+S | 49.2(+1.7) | 70.6(+3.2) | 47.3(+0.6) | 30.4(+16.4) | 130.1(+36.5) | 58.5(+2.0) | 44.9(-0.1) |
| (g) | V+A | V+A | 53.8(+6.3) | 70.9(+3.5) | 47.4(+0.7) | 30.3(+16.3) | 130.7(+37.1) | 62.6(+6.1) | 46.3(+1.3) |
| (h) | V+S | V+S | 49.5(+2.0) | 68.9(+1.5) | 47.4(+0.7) | 42.6(+28.6) | 186.5(+92.9) | 48.2(-8.3) | 43.6(-1.4) |
| (i) | V+A+S | V+A+S | 53.0(+5.5) | 71.7(+4.3) | 47.9(+1.2) | 44.0(+30.0) | 187.5(+93.9) | 58.8(+2.3) | 48.5(+3.5) |
| (j) | V+A& V+S & V+A+S | V+A+S | 55.3(+7.8) | 71.7(+4.3) | 48.2(+1.5) | 45.9(+31.9) | 188.0(+94.4) | 62.2(+5.7) | 48.4(+3.4) |

**A-T Quality.** We train multiple A-T models on different corpora and fine-tune them on audio-text benchmarks, including Clothov2 and AudioCaps. We observe that VALOR-1M and WavCaps are prone to overfitting, so we adjust the training iterations to obtain the best results for models trained on each corpus instead of training for the same number of iterations. As shown in Table 5, model (d) surpasses the baseline and models trained on both VALOR-1M and WavCaps on all four benchmarks.

**OMV-OMC Quality.** We further investigate the correspondence quality between OMV and OMC in the VAST-27M dataset and compare it primarily to the VALOR-1M dataset. All models undergo 50K training steps with $\mathcal{L}_{\text{OM}}$ as the objective, utilizing all modalities in both pretraining and finetuning. The results are presented in Table 6. From the table, we observe that omni-modality pretraining on VAST-27M significantly improves performance across all 7 benchmarks compared to the baseline. Additionally, VAST-27M outperforms VALOR-1M on 5 benchmarks in the MSRVTT and YouCook2 datasets, but lags behind on the VALOR-32K benchmarks due to the consistent video distribution and caption style between VALOR-1M and VALOR-32K. As an additional experiment, we train model (c) on VAST-27M but replace the omni-modality captions generated by LLM with a simple concatenation of vision, audio captions, and subtitles. Model (d) outperforms model (c) on all benchmarks, showcasing the necessity and effectiveness of leveraging the powerful capabilities of LLM to integrate single-modality captions into omni-modality ones.

## 5.4 Ablation Study

We conduct comprehensive ablation studies on pretraining and finetuning models using different modalities for video representations to demonstrate the strength and necessity of omni-modality foundation model pretraining. Models are evaluated on the MSRVTT (open-domain), YouCook2 (subtitle-important), and VALOR-32K (audio-important) benchmarks. The results are presented in Table 7, when no pretraining is applied, models trained with audio and subtitles incorporated (d) can improve the vision-only baseline on all benchmarks. When omni-modality pretraining is applied, the improvement becomes more evidently on most benchmarks, which can be reflected by the comparison between the green values in model (f) and model (d), demonstrating the effectiveness of proposed method. In addition, model (i) outperforms model (f) on all benchmarks, indicating that, similar to the V-T capability, the OMV-OMC capability also benefits from large-scale pretraining. It is worth noting that model (h) and (f) exhibit inferior results on VALOR-32K benchmarks compared to model (g) and even the vision baseline (e). This degradation can be attributed to the inconsistency between pretraining and finetuning, where all clips have subtitles in VAST-27M while most videos in VALOR-32K lack subtitles. When the modality grouping strategy is applied, model (j) demonstrates

good generalization across all types of tasks, as the correlations between text and every modality group have been explored during pretraining.

# 6 Conclusion, Broader Impact and Limitation

In this paper, we introduce the VAST-27M corpus, a large-scale omni-modality video caption dataset, aimed at advancing research in multi-modality pretraining. Each video clip in the dataset is accompanied by automatically generated vision and audio captions, as well as an omni-modality caption that integrates information from vision, audio, and subtitles using a pre-existing Large Language Model (LLM). We also train a unified foundation model called VAST, capable of understanding and connecting the various modalities in videos and captions. Through extensive evaluations, VAST demonstrates its effectiveness in a wide range of vision-text, audio-text, and multi-modal video-text tasks, including retrieval, captioning, and question answering, surpassing existing state-of-the-art methods on public cross-modality benchmarks.

The advancements in multi-modal understanding and generation of video content can have a significant impact on various domains such as entertainment, education, security, transportation, and healthcare, among others. The development of omni-modality foundation models, like VAST-27M and VAST, can contribute to the progress and practical applications in these areas. However, it is important to acknowledge the limitations of our method. For instance, there is a need for more diverse and larger-scale omni-modality corpora beyond VAST-27M. Furthermore, although VAST already supports a wide range of downstream tasks, the integration of LLM is necessary to further enhance its generalization capabilities. Additionally, due to the reason that the data collection process of VAST-27M have used LLM, and the training process of video and audio captioners have utilized some open-sourced cross-modality corpus, so VAST-27M dataset and VAST model may suffer the same bias of those datasets and models.

## Acknowledgments

This work was supported by the National Key Research and Development Program of China (No. 2022ZD0118801), National Natural Science Foundation of China (U21B2043, 62102416, 62206279).

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

# Appendix

## A  More about VAST Foundation Model

### A.1  Pretraining Settings

Specific pretraining configurations of VAST including training corpora, training steps for each corpus (i.e., dataset mix ratio), and training objectives on each corpus are presented in Table 8. To enhance data quality, we use trained vision captioner to generate new captions for CC12M and LAION datasets and replace original captions with them. It is noted that VAST have been trained for relatively small steps (205K steps), but have already shown excellent performances on various types of downstream tasks, and we believe that training by more steps can further increase the model capabilities.

Table 8:  Model configurations and pretraining settings of VAST. It is noted that 400M Web data used in CLIP [40] and LAION-400M [29] used in EVAClip [39] are also counted for training samples statics.  LAION-102M and LAION-110M are both random sampled subsets from LAION-400M. Regarding training objectives, 'ret' represents for the combination of VCC and VCM, while 'cap' denotes VCG, and differnet modality groups are separeted by '%'.

| Model | Param | Sample | Training Corpus | Batch Size | Steps | Epoch | Objectives |
|-------|-------|--------|-----------------|------------|-------|-------|------------|
| VAST | 1.3B | 442M | VAST-27M | 1024 | 60000 | 2.3 | ret%vast%vat%vst%vt%at + cap%vast%vat%vst%vt%at |
| | | | VALOR-1M | 1024 | 25000 | 25 | ret%vat%vt%at + cap%vat%vt%at |
| | | | WavCaps | 1024 | 15000 | 38 | ret%at + cap%at |
| | | | CC4M | 2048 | 30000 | 12 | ret%vt + cap%vt |
| | | | CC12M | 2048 | 20000 | 4 | ret%vt + cap%vt |
| | | | LAION-110M | 2048 | 55000 | 1 | ret%vt + cap%vt |

### A.2  Downstream Datasets Descriptions

We evaluate VAST on multiple popular domnstream datasets, including MSRVTT, VATEX, YouCook2, VALOR-32K, MSVD, LSMDC, DiDeMo, ActivityNet Caption, TGIF, MUSIC-AVQA, TVC, Clotho, AudioCaps, MSCOCO, Flickr30K and VQAv2. Specific train/val/test splits of those benchmarks can be found in Table 9 and specific descriptions of them are as follows.

**MSRVTT** [32] contains 10K video clips and 200K captions.  The videos cover a wide range of topics and scenes, including human activities, sports, natural landscapes, and more. We evaluate text-to-video retrieval, video captioning and video QA on this dataset. Following methods presented in Table 11, we use the '1K-A split' for retrieval evaluation. For captioning and QA, we use the standard split.

**VATEX** [31] contains 41,250 video clips sourced from Kinetics-600 dataset [63] and 825,000 sentence-level descriptions. We evaluate text-to-video retrieval and video captioning on this dataset. For captioning, we use the official split. For retrieval. we follow the HGR [64] split protocol.

**YouCook2** [51] consists of 14K video clips from 2K instructional cooking videos from YouTube. Each video includes multiple actions performed by the chef, along with corresponding textual descriptions and temporal annotations. We evaluate text-to-video retrieval and video captioning on this dataset with official splits.

**VALOR-32K** [7] is an audiovisual video-language benchmark that contains 32K 10 seconds long audible video clips sourced from AudioSet [11]. Each video clip is annotated with an audiovisual caption which simultaneously describes both visual and audio contents in videos.  We evaluate text-to-video retrieval and video captioning on this dataset with official splits.

**MSVD** [33] contains 1,970 videos, each of which is paired with around 40 captions. We evaluate video QA on this dataset and use the split proposed by Xu et al. [44].

**LSMDC** [65] consists of 118K clips form 202 movies, each of which is paird with one caption. We evaluate text-to-video retrieval on this dataset with official split.

Table 9: Downstream dataset splits.

| Task Type | Modal Type | Benchmark | #Videos/#Images | | | #Captions/#QA-pairs | | |
|---|---|---|---|---|---|---|---|---|
| | | | Train | Val | Test | Train | Val | Test |
| Retrieval | V-T(Vis) | MSCOCO | 113287 | 5000 | 5000 | 566747 | 25010 | 25010 |
| | | Flickr30K | 29000 | 1014 | 1000 | 145000 | 5070 | 5000 |
| | A-T | ClothoV1 | 2893 | 1045 | - | 14465 | 5225 | - |
| | | ClothoV2 | 3839 | 1045 | - | 19195 | 5225 | - |
| | | AudioCaps | 49291 | 428 | 816 | 49291 | 2140 | 4080 |
| | V-T(MM) | MSRVTT | 9000 | - | 1000 | 180000 | - | 1000 |
| | | YouCook2 | 10337 | 3492 | - | 10337 | 3492 | - |
| | | VALOR-32K | 25000 | 3500 | 3500 | 25000 | 3500 | 3500 |
| | | VATEX | 25991 | 1500 | 1500 | 259910 | 1500 | 1500 |
| | | DiDeMo | 8394 | 1065 | 1003 | 8394 | 1065 | 1003 |
| | | ANET | 10009 | - | 4917 | 10009 | - | 4917 |
| | | LSMDC | 101046 | 7408 | 1000 | 101046 | 7408 | 1000 |
| Caption | V-T(Vis) | MSCOCO | 113287 | 5000 | 5000 | 566747 | 25010 | 25010 |
| | | MSVD | 1200 | 100 | 670 | 48774 | 4290 | 27763 |
| | A-T | ClothoV1 | 2893 | 1045 | - | 14465 | 5225 | - |
| | | ClothoV2 | 3839 | 1045 | - | 19195 | 5225 | - |
| | | AudioCaps | 49838 | 495 | 975 | 49438 | 2475 | 4875 |
| | V-T(MM) | MSRVTT | 6513 | 497 | 2990 | 130260 | 9940 | 59800 |
| | | YouCook2 | 10337 | 3492 | - | 10337 | 3492 | - |
| | | VALOR-32K | 25000 | 3500 | 3500 | 25000 | 3500 | 3500 |
| | | VATEX | 25991 | 3000 | 6000 | 259910 | 30000 | 60000 |
| | | TVC | 86603 | 10841 | - | 174350 | 43580 | - |
| QA | V-T(Vis) | MSVD-QA | 1200 | 250 | 520 | 30933 | 6,415 | 13157 |
| | | TGIF-FrameQA | 32345 | - | 7132 | 39389 | - | 13691 |
| | | VQAv2 | 82783 | 40504 | 37K/81K | 4437570 | 2143540 | 1.1M/4.5M |
| | V-T(MM) | MSRVTT-QA | 6513 | 497 | 2990 | 158581 | 12278 | 72821 |
| | | MUSIC-AVQA | 9277 | 3815 | 6399 | 32087 | 4595 | 9185 |
| | | ANET-QA | 3200 | 1800 | 800 | 32000 | 18000 | 8000 |

**DiDeMo** [52] contains 10K long-form videos from Flickr and for each video, four short sentences are annotated in temporal order. We follow methods in Table 11 to concatenate those short sentences and evaluate 'paragraph-to-video' retrieval on this benchmark. The official split is used.

**ActivityNet Caption** [53] contains 20K long-form videos (180s as average length) from YouTube and 100K captions. We evaluate text-to-video retrieval and video QA on this dataset. For retrieval we use official split and for video QA, split proposed by Yu et al. [60] is used.

**TGIF** [66] contains three video QA benchmarks including TGIF-Action, TGIF-transition and TGIF-Frame, and the first two are multiple-choice QA while the last is open-ended QA. We evaluate VAST on TGIF-frame benchmark with official split.

**MUSIC-AVQA** [59] is a audiovisual video QA benchmark containing more than 45K Q-A pairs covering 33 different question templates spanning over different modalities and question types. The offcial split is used.

**TVC** [58] is a multi-channel video captioning dataset containing 108K video moments and 262K paired captions. Video subtitles can be used as additional input. We evaluate video captioning on this benchmark with official split.

**Clotho** [12] contains 15-30 second audio clips and has two versions. The original (v1) has 4981 audios, while an expanded version (v2) includes 6974 audios, enlarging solely the training set. We evaluate text-to-audio retrieval and audio captioning on those benchmarks with official split.

**AudioCaps** [13] contains 51K 10-second clips, with one caption in the training set and five in the validation and test sets. We evaluate text-to-audio retrieval and audio captioning on it. For captioning, we use the official split, and for retrieval we follow the Sophia et al. [67] split protocol.

**MSCOCO** [30] contains 123K images each of which is paired with 5 annotated captions, We evaluate text-to-image retrieval and image captioning on this dataset with Karpathy split [68].

Table 10: Downstream task finetuning settings. Lr, Bs, Epo, Obj and Res denote learning rate, batch size, epoch, training objectives and resolution, respectively. Vf(Tr), Vf(Te), Ac(Tr), Ac(Te) denotes sampled video frames (Vf) or audio clips (Ac) in training (Tr) and testing (Te), respectively. The marks in Obj are the same as those in Table 8. Most hyperparameters in the table are not precisely tuned.

| Task | Modality | Benchmark | Lr | Bs | Epo | Obj | Vf(Tr) | Vf(Te) | Ac(Tr) | Ac(Te) | Res |
|---|---|---|---|---|---|---|---|---|---|---|---|
| RET | V-T(Vis) | MSCOCO | 1e-5 | 256 | 5 | ret%vt | - | - | - | - | 384 |
| | | Flickr | 1e-5 | 256 | 5 | ret%vt | - | - | - | - | 384 |
| | A-T | ClothoV1/V2 | 2e-5 | 64 | 10 | ret%at | - | - | 3 | 3 | - |
| | | AudioCaps | 2e-5 | 64 | 10 | ret%at | - | - | 1 | 1 | - |
| | V-T(MM) | MSRVTT | 2e-5 | 64 | 3.6 | ret%vast | 8 | 16 | 1 | 1 | 224 |
| | | YouCook2 | 3e-5 | 64 | 30 | ret%vast | 8 | 16 | 1 | 1 | 224 |
| | | VALOR-32K | 2e-5 | 64 | 10 | ret%vat | 8 | 8 | 1 | 1 | 224 |
| | | VATEX | 2e-5 | 64 | 2.5 | ret%vast | 8 | 16 | 1 | 1 | 224 |
| | | DiDeMo | 2e-5 | 64 | 40 | ret%vat | 8 | 32 | 2 | 2 | 224 |
| | | ANET | 2e-5 | 64 | 20 | ret%vat | 8 | 32 | 2 | 2 | 224 |
| | | LSMDC | 2e-5 | 64 | 5 | ret%vat | 8 | 32 | 1 | 1 | 224 |
| CAP | V-T(Vis) | MSCOCO | 1e-5 | 64 | 5 | cap%vt | - | - | - | - | 480 |
| | | MSCOCO(SCST) | 2.5e-6 | 64 | 2.5 | cap%vt | - | - | - | - | 480 |
| | A-T | ClothoV1/V2 | 2e-5 | 64 | 10 | cap%at | - | - | 3 | 3 | - |
| | | AudioCaps | 2e-5 | 64 | 10 | cap%at | - | - | 1 | 1 | - |
| | V-T(MM) | MSRVTT | 2e-5 | 128 | 10 | cap%vast | 8 | 8 | 1 | 1 | 224 |
| | | YouCook2 | 3e-5 | 64 | 30 | cap%vast | 8 | 16 | 1 | 1 | 224 |
| | | VALOR-32K | 1e-5 | 64 | 10 | cap%vat | 8 | 12 | 1 | 1 | 224 |
| | | VATEX | 2e-5 | 64 | 10 | cap%vast | 8 | 20 | 1 | 1 | 224 |
| | | VATEX(SCST) | 7e-6 | 64 | 5 | cap%vast | 8 | 20 | 1 | 1 | 224 |
| | | TVC | 3e-5 | 64 | 40 | cap%vst | 8 | 8 | - | - | 224 |
| QA | V-T(Vis) | MSVD-QA | 1e-5 | 64 | 10 | qa%vt | 8 | 14 | - | - | 224 |
| | | TGIF-FrameQA | 2e-5 | 64 | 10 | qa%vt | 4 | 4 | - | - | 224 |
| | | VQAv2 | 2e-5 | 128 | 20 | qa%vt | - | - | - | - | 384 |
| | V-T(MM) | MSRVTT-QA | 2e-5 | 64 | 4.5 | qa%vast | 8 | 8 | 1 | 1 | 224 |
| | | MUSIC-AVQA | 2e-5 | 64 | 20 | qa%vat | 8 | 8 | 2 | 2 | 224 |
| | | ANET-QA | 2e-5 | 64 | 10 | qa%vat | 8 | 16 | 2 | 2 | 224 |

**Flickr30K** [42] contains 31K images each of which is paired with 5 annotated captions, We evaluate text-to-image retrieval on this dataset with Karpathy split [68].

**VQAv2** [45] was used as the basis of the 2017 VQA Challenge2, it contains 1.1M questions with 11.1M answers relating to MSCOCO images. The official split is used.

### A.3    Finetuning Settings

Specific finetuning hyperparameters of VAST for different benchmarks are presented in Table 10.

### A.4    Detailed Comparisons to State-of-the-Art Methods

**Text-to-Video Retrieval**. We compare VAST to SOTA methods on six multi-modal text-to-video retrieval benchmarks. As shown in Table 4, VAST improves previous SOTA methods by 5.1, 1.6, 3.7, 6.1 points on MSRVTT, DiDeMo, ActivityNet, VATEX benchmarks, respectively. Besides above mentioned vision-oriented benchmarks, VAST outperforms VALOR-L [7] by 6.8 points on the audio-oriented benchmark VALOR-32K, and surpass MELTR [55] by 16.7 points on the subtitle-oriented benchmark YouCook2, which demonstrate the strong generalization capabilities of VAST towards different types of downstream datasets. In addition, the zero-shot retrieval performance comparison is shown in Table 12, VAST achieves 49.3 and 55.5 zero-shot R@1 performance that surpasses previous SOTA by 7.3 and 6.9 points, respectively.

**Video QA**. We evaluate VAST on five open-ended video QA benchmarks. As shown in Table 13, VAST have achieved new SOTA performances on all benchmarks, and outperform recent proposed large-scale foundation models such as GIT [19], MaMMUT [49], Flamingo [92] and CoCa [93]. In addition, on the audiovisual video QA benchmark MUSIC-AVQA, VAST surpasses VALOR by 1.8 points, demonstrating its better capabilities to answer both visual and audio questions.

Table 11: Performance comparison on Text-to-Video Retrieval benchmarks. Recall@1,5,10 are used as evaluation metrics. For fair comparisons, performances before employing post-processing such as dual-softmax [69] are reported and compared. All benchmarks are multi-modal benchmarks (containing audio and subtitle tracks). Methods utilizing audio or subtitle modalities besides vision for video representation are marked with gray background color.

| Method | Sample | MSRVTT | | | DiDeMo | | | ActivityNet | | |
|---|---|---|---|---|---|---|---|---|---|---|
| | | R@1 | R@5 | R@10 | R@1 | R@5 | R@10 | R@1 | R@5 | R@10 |
| Singularity [70] | 17M | 41.5 | 68.7 | 77.0 | 53.9 | 79.4 | 86.9 | 47.1 | 75.5 | 85.5 |
| OmniVL [71] | 17M | 47.8 | 74.2 | 83.8 | 52.4 | 79.5 | 85.4 | - | - | - |
| HiTeA [72] | 17M | 46.8 | 71.2 | 81.9 | 56.5 | 81.7 | 89.7 | 49.7 | 77.1 | 86.7 |
| VINDLU-L [73] | 25M | 48.8 | 72.4 | 82.2 | 59.8 | 86.6 | 91.5 | 55.9 | 82.3 | 90.9 |
| LAVENDER [74] | 30M | 40.7 | 66.9 | 77.6 | 53.4 | 78.6 | 85.3 | - | - | - |
| All-in-one [75] | 138M | 37.9 | 68.1 | 77.1 | 32.7 | 61.4 | 73.5 | - | - | - |
| CLIP4Clip [56] | 400M | 44.5 | 71.4 | 81.6 | 43.4 | 70.2 | 80.6 | 40.5 | 72.4 | - |
| X-CLIP [76] | 400M | 49.3 | 75.8 | 84.8 | 47.8 | 79.3 | - | 46.2 | 75.5 | - |
| mPLUG-2 [77] | 417M | 53.1 | 77.6 | 84.7 | 56.4 | 79.1 | 85.2 | - | - | - |
| UMT-L [54] | 425M | 58.8 | 81.0 | 87.1 | 70.4 | **90.1** | **93.5** | 66.8 | 89.1 | 94.9 |
| CLIP-VIP [78] | 500M | 54.2 | 77.2 | 84.8 | 50.5 | 78.4 | 87.1 | 53.4 | 81.4 | 90.0 |
| MMT [21] | 136M | 26.6 | 57.1 | 69.6 | - | - | - | 28.7 | 61.4 | - |
| AVLNet [25] | 136M | 22.5 | 50.5 | 64.1 | - | - | - | - | - | - |
| Gabeur et al. [79] | 136M | 28.7 | 59.5 | 70.3 | - | - | - | 29.0 | 61.7 | - |
| ECLIPSE [80] | 400M | - | - | - | 44.2 | - | - | 45.3 | 75.7 | 86.2 |
| VALOR-L [7] | 433.5M | 54.4 | 79.8 | 87.6 | 57.6 | 83.3 | 88.8 | 63.4 | 87.8 | 94.1 |
| **VAST** | **442M** | **63.9** | **84.3** | **89.6** | **72.0** | 89.0 | 91.4 | **70.5** | **90.9** | **95.5** |

| Method | VATEX | | | Method | VALOR-32K | | | Method | YouCook2 | | |
|---|---|---|---|---|---|---|---|---|---|---|---|
| | R@1 | R@5 | R@10 | | R@1 | R@5 | R@10 | | R@1 | R@5 | R@10 |
| Support-set [81] | 44.9 | 82.1 | 89.7 | Frozen [3] | 32.9 | 60.4 | 71.2 | UniVL [23] | 28.9 | 57.6 | 70.0 |
| CLIP4Clip [56] | 55.9 | 89.2 | 95.0 | CLIP4Clip [56] | 43.4 | 69.9 | 79.7 | MELTR [55] | 33.7 | 63.1 | 74.8 |
| DCR [82] | 65.7 | 92.6 | 96.7 | | | | | VLM [83] | 27.1 | 56.9 | 69.4 |
| VALOR-L | 76.9 | 96.7 | 98.6 | AVLNet [25] | 21.6 | 47.2 | 59.8 | VALUE [20] | 31.3 | 53.0 | 62.2 |
| **VAST** | **83.0** | **98.2** | **99.2** | VALOR-L [7] | 73.2 | 91.6 | 95.4 | **VAST** | **50.4** | **74.3** | **80.8** |
| | | | | **VAST** | **80.0** | **93.7** | **96.6** | | | | |

Table 12: Performance comparison on zero-shot Text-to-Video Retrieval benchmarks. Methods utilizing audio or subtitle modalities besides vision for video representation are marked with gray background color.

| Method | Sample | MSRVTT | | | DiDeMo | | |
|---|---|---|---|---|---|---|---|
| | | R@1 | R@5 | R@10 | R@1 | R@5 | R@10 |
| Frozen [3] | 5M | 18.7 | 39.5 | 51.6 | 21.1 | 46.0 | 56.2 |
| ALPRO [84] | 5M | 24.1 | 44.7 | 55.4 | 23.8 | 47.3 | 57.9 |
| Singularity [70] | 5M | 28.4 | 50.2 | 59.5 | 36.9 | 61.6 | 69.3 |
| HiTeA [72] | 17M | 34.4 | 60.0 | 69.9 | 43.2 | 69.3 | 79.0 |
| OmniVL [71] | 18M | 42.0 | 63.0 | 73.0 | 40.6 | 64.6 | 74.3 |
| VIOLET [85] | 183M | 25.9 | 49.5 | 59.7 | 23.5 | 49.8 | 59.8 |
| UMT-L [54] | 425M | 40.7 | 63.4 | 71.8 | 48.6 | 72.9 | 79.0 |
| Florence [86] | 900M | 37.6 | 63.8 | 72.6 | - | - | - |
| **VAST** | **443M** | **49.3** | **68.3** | **73.9** | **55.5** | **74.3** | **79.6** |

**Video Captioning**. In Table 14, we compare VAST to state-of-the-art methods on five multi-modal video captioning benchmarks. According to the results, VAST have achieved new state-of-the-art CIDEr score on all five benchmarks with evident margins. Compared to previous vision-language modal SOTA method GIT [19] which takes a 5.1B DaViT [100] as vision encoder and conduct pretraining on 12.9B private image-text corpus, VAST surpass it with only 22.5% parameters and 3.4% training data, demonstrating the high efficiency of our method. Compared to previous multi-modal video-language SOTA method VALOR [19], VAST can additionally process subtitle-oriented benchmarks such as YouCook2 and TVC, and achieves better results due to that it jointly models the relations between text and omni-modalities in videos.

Table 13: Performance comparison on Video QA benchmarks. Acc is used as evaluation metric. MSVD-QA and TGIF-QA are vision-only benchmarks while the others are multi-modal benchmarks. Methods utilizing audio or subtitle modalities besides vision for video representation are marked with gray background color.

| Method | Sample | MSRVTT-QA | MSVD-QA | TGIF-QA | ActivityNet-QA | MUSIC-AVQA |
|---|---|---|---|---|---|---|
| ClipBERT [87] | 5.4M | 37.4 | - | 60.3 | | - |
| ALPRO [84] | 5M | 42.1 | 45.9 | - | - | - |
| VIOLETv2 [88] | 5M | 44.5 | 54.7 | 72.8 | - | - |
| Clover [89] | 5M | 43.9 | 51.9 | 71.4 | - | - |
| OmniVL [71] | 17M | 44.1 | 51.0 | | | - |
| HiTeA [72] | 17M | 45.9 | 55.3 | 73.2 | 46.4 | - |
| SINGULARITY [70] | 17M | 43.5 | - | - | 43.1 | - |
| VINDLU-B [73] | 17M | 43.8 | - | - | 44.6 | - |
| LAVENDER [74] | 30M | 45.0 | 56.6 | 73.5 | - | - |
| JustAsk [90] | 69M | 41.5 | 46.3 | - | 38.9 | - |
| MERLOT [6] | 180M | 43.1 | - | 69.5 | 41.4 | - |
| All-in-one [75] | 228.5M | 46.8 | 48.3 | 66.3 | - | - |
| FrozenBiLM [91] | 410M | 47.0 | 54.8 | 68.6 | 43.2 | - |
| mPLUG-2 [77] | 417M | 48.0 | 58.1 | 75.4 | - | - |
| UMT-L [54] | 425M | 47.1 | 55.2 | - | - | - |
| InternVideo [57] | 646M | 47.1 | 55.5 | 72.2 | - | - |
| GIT [19] | 1.7B | 43.2 | 56.8 | 72.8 | - | - |
| MaMMUT [49] | 2B | 49.5 | 60.2 | - | - | - |
| Flamingo (80B) [92] | 2.3B | 47.4 | - | - | - | - |
| VideoCoCa (2.1B) [93] | 4.8B | 46.0 | 56.9 | - | - | - |
| GIT2 (5.1B) [19] | 12.9B | 45.6 | 58.2 | 74.9 | - | - |
| VALOR-L [7] | 433.5M | 49.2 | 60.0 | 78.7 | 48.6 | 78.9 |
| **VAST(1.3B)** | **442M** | **50.1** | **60.2** | **79.1** | **50.4** | **80.7** |

Table 14: Performance comparison on Video Captioning benchmarks. All benchmarks are multi-modal benchmarks. BLEU@4 (B@4) and CIDEr (C) metrics are used as evaluation metrics. On VATEX benchmark, we follow most state-of-the-art methods [19; 7; 62] employing SCST finetuning [41] after cross-entropy training, and corresponding results are marked with '*'. Methods utilizing audio or subtitle modalities besides vision for video representation are marked with gray background color.

| Method | Sample | MSRVTT | | VATEX | | YouCook2 | | TVC | | VALOR-32K | |
|---|---|---|---|---|---|---|---|---|---|---|---|
| | | B@4 | C | B@4 | C | B@4 | C | B@4 | C | B@4 | C |
| SwinBERT [61] | - | 41.9 | 53.8 | 38.7 | 73.0 | 9.0 | 109.0 | 14.5 | 55.4 | 5.4 | 27.3 |
| VIOLETv2 [88] | 5M | - | 58.0 | - | - | - | - | - | - | - | - |
| HiTeA [72] | 5M | - | 62.5 | - | - | - | - | - | - | - | - |
| LAVENDER [74] | 30M | - | 60.1 | - | - | - | - | - | - | - | - |
| MaMMUT [49] | 2B | - | 73.6 | - | - | - | - | - | - | - | - |
| GIT [19] | 1.7B | 53.8 | 73.9 | 41.6* | 91.5* | 10.3 | 129.8 | 16.2 | 63.0 | | |
| GIT2(5.1B) [19] | 12.9B | 54.8 | 75.9 | 42.7* | 94.5* | 9.4 | 131.2 | 16.9 | 66.1 | - | - |
| SMPFF [22] | - | 48.4 | 58.5 | 39.7 | 70.5 | - | - | - | - | 7.5 | 37.1 |
| VALUE [20] | 136M | - | - | - | 58.1 | 12.4 | 130.3 | 11.6 | 50.5 | - | - |
| UniVL [23] | 136M | 41.8 | 50.0 | - | - | 17.4 | 181.0 | - | - | - | - |
| MELTR [55] | 136M | 44.2 | 52.8 | - | - | 17.9 | 190.0 | - | - | - | - |
| CLIP4Caption++ [62] | 400M | - | - | 40.6* | 85.7* | - | - | 15.0 | 66.0 | - | - |
| VALOR-L [7] | 433.5M | 54.4 | 74.0 | **45.6*** | 95.8* | - | - | - | - | 9.6 | 61.5 |
| **VAST(1.3B)** | **442M** | **56.7** | **78.0** | 45.0* | **99.5*** | **18.2** | **198.8** | **19.9** | **74.1** | **9.9** | **62.2** |

Table 15: Performance comparison on Text-to-Audio Retrieval benchmarks. Recall@1,5,10 are used as evaluation metrics.

| Method | ClothoV1 | | | ClothoV2 | | | AudioCaps | | |
|---|---|---|---|---|---|---|---|---|---|
| | R@1 | R@5 | R@10 | R@1 | R@5 | R@10 | R@1 | R@5 | R@10 |
| Oncescu et al. [67] | 9.6 | - | 40.1 | - | - | - | 25.1 | - | 73.2 |
| Nagrani et al. [94] | 12.6 | - | 45.4 | - | - | - | 35.5 | - | 84.5 |
| LAION [16] | - | - | - | 16.1 | 38.3 | 51.1 | 36.1 | 71.8 | 83.9 |
| CNN14-BERT [18] | - | - | - | 21.5 | 47.9 | 61.9 | 35.1 | 70.0 | 82.1 |
| HTSAT-BERT [18] | - | | - | 19.7 | 45.7 | 59.4 | 42.2 | 76.5 | 87.1 |
| VALOR-B [7] | 17.5 | 42.7 | 55.3 | - | - | - | 40.1 | 73.9 | 83.1 |
| ONE-PIECE [95] | - | - | - | 22.4 | 49.0 | 62.7 | 42.5 | **77.5** | **88.4** |
| **VAST** | **25.1** | **51.5** | **64.0** | **26.9** | **53.2** | **66.1** | **52.0** | 76.8 | 82.9 |

Table 16: Performance comparison on Audio Captioning benchmarks. BLEU@4 (B@4) and CIDEr (C) metrics are used as evaluation metrics.

| Method | ClothoV1 | | | | ClothoV2 | | | | AudioCaps | | | |
|---|---|---|---|---|---|---|---|---|---|---|---|---|
| | B@4 | M | R | C | B@4 | M | R | C | B@4 | M | R | C |
| Xu et al. [96] | 15.9 | 16.9 | 36.8 | 37.7 | - | - | - | - | 23.1 | 22.9 | 46.7 | 66.0 |
| CNN14-BART [18] | - | - | - | - | 18.0 | 18.5 | 40.0 | 48.8 | 27.2 | 24.7 | 49.9 | 75.6 |
| HTSAT-BART [18] | - | - | - | - | 16.8 | 18.4 | 38.3 | 46.2 | 28.3 | **25.0** | 50.7 | **78.7** |
| VALOR-B [7] | 16.2 | 17.4 | 38.2 | 42.3 | - | - | - | - | 27.0 | 23.1 | 49.4 | 74.1 |
| **VAST** | **18.5** | **18.9** | **39.9** | **50.7** | **19.0** | **19.3** | **40.8** | **51.9** | **29.5** | 24.7 | **50.9** | 78.1 |

**Text-to-Audio Retrieval and Audio Captioning**. As shown in Table 15, VAST have largely improved previous SOTA methods on three text-to-audio retrieval benchmarks, by 7.6, 5.4 and 9.8 R@1 points, respectively. and for audio captioning task, VAST achieves new SOTA performances on Clotho benchmark (both V1 and V2), and comparable performance on AudioCaps benchmark to WavCaps [18]. It is noted that WavCaps explored four model architectures with different audio encoder and text encoders targeting at different benchmarks, while VAST takes a unified architecture without targeted optimizations for specific downstream benchmarks.

**Image-Text Benchmarks**. We evaluate VAST on text-to-image retrieval, image captioning and image QA benchmarks. The results are presented in Table 17, from which we can find that even though VAST is designed as a omni-modality video-language understanding and generation model, it also shows strong capabilities on image-text benchmarks, demonstrating its generalization capabilities towards tasks of various modality types. Specifically, VAST achieves new SOTA performance on R@1

Table 17: Performance comparison on Image-Text downstream tasks. CIDEr (C) and SPICE (S) metrics are reported for captioning. On MSCOCO caption benchmark, we follow SOTA methods [19; 7; 48] employing SCST finetuning [41], and corresponding results are marked with '*'.

| Method | Sample | MSCOCO-Ret | | | Flickr30K-Ret | | | MSCOCO-Cap | | VQAv2 | |
|---|---|---|---|---|---|---|---|---|---|---|---|
| | | R@1 | R@5 | R@10 | R@1 | R@5 | R@10 | C | S | dev | std |
| ALBEF [38] | 14M | 60.7 | 84.3 | 90.5 | 85.6 | 97.5 | 98.9 | - | - | 75.84 | 76.04 |
| OFA [48] | 18M | - | - | - | - | - | - | **154.9*** | 26.6* | 82.0 | 82.0 |
| BEiT-3 [47] | 21M | 67.2 | 87.7 | 92.8 | 90.3 | **98.7** | 99.5 | 147.6 | 25.4 | 84.19 | 84.03 |
| BLIP [28] | 129M | 65.1 | 86.3 | 91.8 | 87.6 | 97.7 | 99.0 | 136.7 | - | 78.25 | 78.32 |
| BLIP-2 [46] | 129M | **68.3** | 87.7 | 92.6 | - | - | - | 145.8 | - | 82.19 | 82.30 |
| mPLUG-2 [77] | 417M | 65.7 | 87.1 | 92.6 | 88.1 | 97.6 | 99 .1 | 137.7 | 23.7 | 81.11 | 81.13 |
| VALOR-L [7] | 433.5M | 61.4 | 84.4 | 90.9 | - | - | - | 152.5* | 25.7* | 78.46 | 78.62 |
| Florence [86] | 900M | 63.2 | 85.7 | - | 87.9 | 98.1 | - | - | - | 80.16 | 80.36 |
| PaLI [50] | 1.6B | - | - | - | - | - | - | 149.1 | - | **84.3** | **84.3** |
| GIT [19] | 1.7B | - | - | - | - | - | - | 151.1* | 26.3* | 78.6 | 78.8 |
| SimVLM [97] | 1.8B | - | - | - | - | - | - | 143.3 | 25.4 | 80.03 | 80.34 |
| ALIGN [98] | 1.8B | 59.9 | 83.3 | 89.8 | 84.9 | 97.4 | 98.6 | - | - | - | - |
| Flamingo (80B) [92] | 2.3B | - | - | - | - | - | - | 138.1 | - | 82.0 | 82.1 |
| CoCa(2.1B) [99] | 4.8B | - | - | - | - | - | - | 143.6 | 24.7 | 82.3 | 82.3 |
| GIT2(5.1B) [19] | 12.9B | - | - | - | - | - | - | 152.7* | 26.4* | 81.7 | 81.9 |
| **VAST** | 442M | 68.0 | **87.7** | **92.8** | **91.0** | 98.5 | **99.5** | 149.0* | **27.0*** | 80.23 | 80.19 |

score of Flicker30K and R@5, R@10 scores of MSCOCO dataset, which outperforms image-text pretrained foundation models such as BLIP-2 [46] and BEiT-3 [47]. On COCO caption benchmark, VAST achieves 27.0 SPICE score which outperforms all previous methods such as OFA [48] and GIT2 [19]. On image QA benchmark, VAST achieves better performance than GIT [19], which is also a generative methods and predicts answers in a fully open way without any constraints.

# B  More about VAST-27M Dataset

## B.1  Word cloud distribution

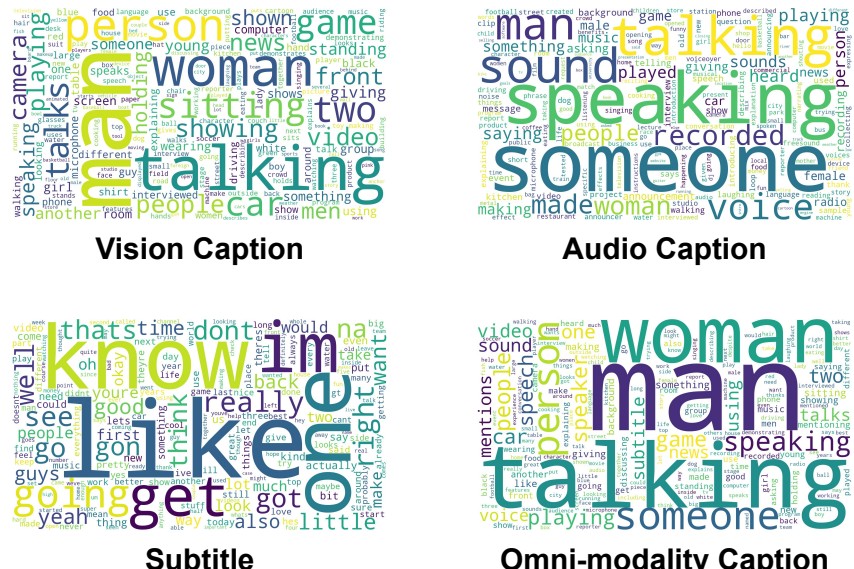

Figure 3: Word cloud map (Top-200) for vision, audio, omni-modality captions and raw subtitles of VAST-27M.

## B.2  Prompts for Omni-Modality Caption Generation

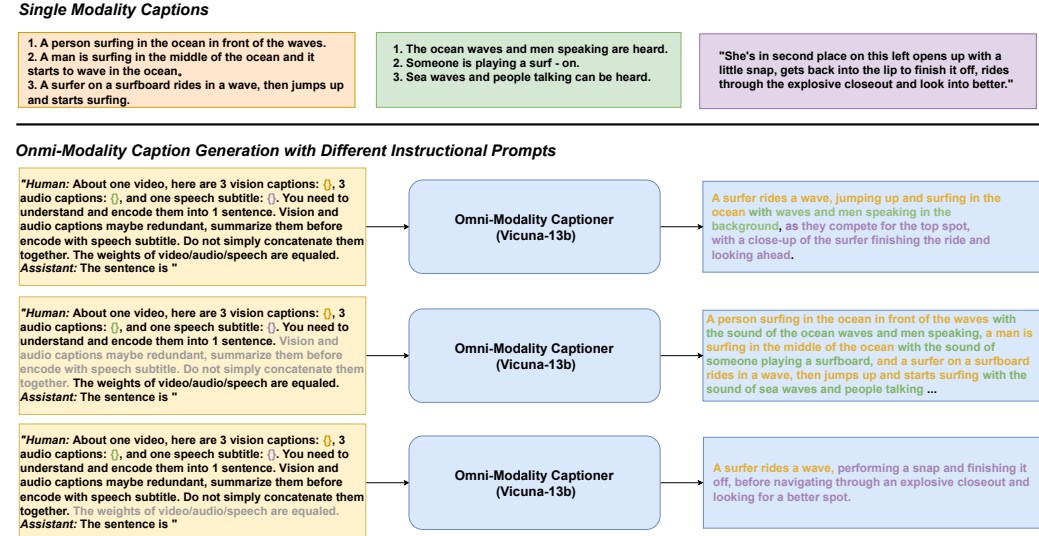

Figure 4: Ablation study for instructional prompt used for omni-modality video caption generation in VAST-27M.

## B.3    More Examples

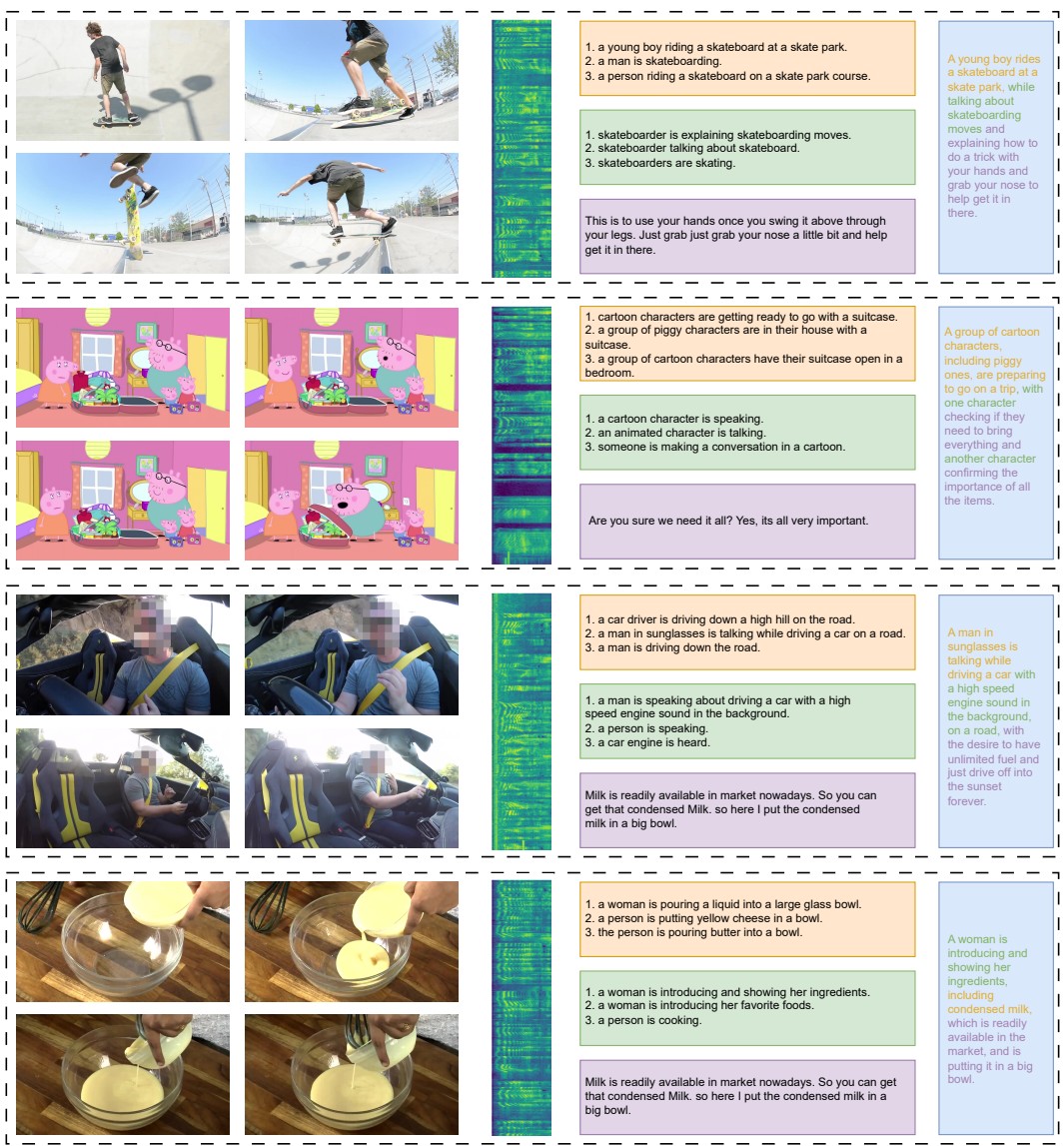

Figure 5: More samples in VAST-27M.

