|---|---|---|---|
| | R@1 | R@5 | R@10 |
| Support-set [43] | 44.9 | 82.1 | 89.7 |
| CLIP4Clip [34] | 55.9 | 89.2 | 95.0 |
| DCR [44] | 65.7 | 92.6 | 96.7 |
| VALOR-L | 76.9 | 96.7 | 98.6 |
| **VAST** | **83.0** | **98.2** | **99.2** |

| Method | VALOR-32K | | |
|---|---|---|---|
| | R@1 | R@5 | R@10 |
| Frozen [45] | 32.9 | 60.4 | 71.2 |
| CLIP4Clip [34] | 43.4 | 69.9 | 79.7 |
| AVLNet [40] | 21.6 | 47.2 | 59.8 |
| VALOR-L [9] | 73.2 | 91.6 | 95.4 |
| **VAST** | **80.0** | **93.7** | **96.6** |

| Method | YouCook2 | | |
|---|---|---|---|
| | R@1 | R@5 | R@10 |
| UniVL [46] | 28.9 | 57.6 | 70.0 |
| MELTR [47] | 33.7 | 63.1 | 74.8 |
| VLM [48] | 27.1 | 56.9 | 69.4 |
| VALUE [49] | 31.3 | 53.0 | 62.2 |
| **VAST** | **50.4** | **74.3** | **80.8** |

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

Table 8: Performance comparison on Text-to-Audio Retrieval benchmarks.

| Method | Sample | ClothoV1 | | | ClothoV2 | | | AudioCaps | | |
|---|---|---|---|---|---|---|---|---|---|---|
| | | R@1 | R@5 | R@10 | R@1 | R@5 | R@10 | R@1 | R@5 | R@10 |
| Oncescu et al. [22] | - | 9.6 | - | 40.1 | - | - | - | 25.1 | - | 73.2 |
| Nagrani et al. [68] | 1M | 12.6 | - | 45.4 | - | - | - | 35.5 | - | 84.5 |
| LAION [69] | 0.63M | - | - | - | 16.1 | 38.3 | 51.1 | 36.1 | 71.8 | 83.9 |
| CNN14-BERT [70] | 0.4M | - | - | - | 21.5 | 47.9 | 61.9 | 35.1 | 70.0 | 82.1 |
| HTSAT-BERT [70] | 0.4M | - | - | - | 19.7 | 45.7 | 59.4 | 42.2 | 76.5 | **87.1** |
| VALOR-B [9] | 1M | 17.5 | 42.7 | 55.3 | - | - | - | 40.1 | 73.9 | 83.1 |
| **VAST** | **28.4M** | **25.1** | **51.5** | **64.0** | **26.9** | **53.2** | **66.1** | **52.0** | **76.8** | 82.9 |

Table 9: Performance comparison on Audio Captioning benchmarks.

| Method | Sample | ClothoV1 | | | | ClothoV2 | | | | AudioCaps | | | |
|---|---|---|---|---|---|---|---|---|---|---|---|---|---|
| | | B@4 | M | R | C | B@4 | M | R | C | B@4 | M | R | C |
| Xu et al. [71] | - | 15.9 | 16.9 | 36.8 | 37.7 | - | - | - | - | 23.1 | 22.9 | 46.7 | 66.0 |
| CNN14-BART [70] | 0.4M | - | - | - | - | 18.0 | 18.5 | 40.0 | 48.8 | 27.2 | 24.7 | 49.9 | 75.6 |
| HTSAT-BART [70] | 0.4M | - | - | - | - | 16.8 | 18.4 | 38.3 | 46.2 | 28.3 | **25.0** | 50.7 | **78.7** |
| VALOR-B [9] | 1M | 16.2 | 17.4 | 38.2 | 42.3 | - | - | - | - | 27.0 | 23.1 | 49.4 | 74.1 |
| **VAST** | **28.4M** | **18.5** | **18.9** | **39.9** | **50.7** | **19.0** | **19.3** | **40.8** | **51.9** | **29.5** | 24.7 | **50.9** | 78.1 |