# OpenReview forum: "VAST: A Vision-Audio-Subtitle-Text Omni-Modality Foundation Model and Dataset"
_NeurIPS.cc/2023/Conference — NeurIPS 2023 poster_

### Official Review · Reviewer_zFyv · 2023-07-01

**Soundness:** 2 fair
**Presentation:** 1 poor
**Contribution:** 2 fair
**Rating:** 3
**Confidence:** 4

**Summary:**

This paper proposes an automatic pipeline to create captions for videos, where multiple modalities are involved in the generation process of the captions. Authors initially train separate audio and video-based captioners. Then, those audiovisual-driven captions, alongside subtitles are restructured into a single caption by prompting a pertained LLM that is responsible for aggregating multiple text/caption sources. Next, such a dataset is used to train a multimodal model capable of addressing a variety of text-to-video/audio retrieval, generation, and question-answering tasks.

**Strengths:**

1. This work confirms that today’s unimodal caption generation models (datasets) are very strong baselines (resources) that diminish the need for a multimodal data collection process which could be costly and non-scaleable, thanks to summarization and aggregation capabilities of the latest LLMs.
2. The downstream experiments are comprehensive and show promising results.


**Weaknesses:**

1. Paper is not easy to follow, especially due to the poor choice of naming for components (i.e features, losses etc) and abbreviations.
2. Paper is highly incremental. It is heavily built on top of numerous existing pertained models and offers almost no technical novelty even in the aggregation step.
3. Authors propose multiple components to the loss function (Eq 1-5). Still, there is almost no discussion on how these different components interact with one another, and impact the quality of learned representations. The experimental section is only focused on downstream tasks and the generated dataset.
4. The dataset generation step is shaped by the power of the pertained models which authors have adopted. They are also combining multiple datasets during pertaining. Hence, outperforming SOTA is not only expected but one wonders whether such comparisons are even fair. At the end of the day, VAST stands on the shoulders of numerous pertained models developed by others.

**Questions:**

1. On OM-VCM: $f_{omv}$ which is fed to the text encoder plays the role of condition, however, a part of  $f_{omv}$ is $f_{s}$ which itself is the output of text encoder. I am not quite sure how this makes sense in terms of the space of inputs to the text encoder. The authors should clarify and elaborate.
2. Line 206: what does “we uniformly model the relations of V-T, A-T, VA-T, VS-T, and VAS-T ” mean?
3. Eq 2: why this cannot be captured via contrastive loss? Is it only due to maintaining temporal resolution? If so, authors should try sequences of length 1 which is [cls] token to confirm that.
4. Line 219: using a single video frame is quite strange. Considering the length of video clips (e.g 5-30 seconds for VAST) does it make sense to pick a single frame, especially if we are aiming to create captions for videos? Did authors try more than 1 frame during pertaining to study the effect of such choice?
5. Line 223-224: What is the contribution of VCM. In other words, what if you equally weigh the losses or reverse them, how do results vary?


**Limitations:**

This work relies on multiple pretrained models and datasets. The bias-fairness concerns hence scale considerably due to such a design. On top of it is the choice of leveraging LLMs whose hidden biases are not quite yet clear for the research community. This is not to discourage using such models, but I believe authors can further improve the limitation and broader impact section of the paper.

---

> ### Author Rebuttal · Authors · 2023-08-08
>
> **weakness 1**
> Considering that the motivation of our work is to build a more general comprehensive omni-modality foundation model,  it unavoidably involves multiple modalities (text, audio, vision, subtitle), multiple features (single/fusion modality features, global/local features), multiple losses (contrastive, matching, captioning). We have indeed paid efforts to make those concepts clear but considering the complexity of algorithms themselves, perhaps the representations seems a little bit complicated and takes some time for understanding, we are sorry about this and will further improve representations in next version.
>
> **weakness 2**
> + Firstly, the technical novalty has been illustrated in the global rebuttal [Common Question 1].
> + Secondly, we think that the statement about "VAST is highly incremental because the single-modality encoders are initialized by other models" is not appropriate.  The target of cross-modality pretraining research is to build strong connections between modalities and serve as a miultimodal generlist for downstream tasks like captioning and retrieval, rather than training single modality representations from scratch to serve for single-modality tasks. Likewisely, in CV filed most segmentation and detection methods are developed on the basis of vision backbone (ResNet or ViT), and it's not appropriate to say that all those methods are incremental.
> + Thirdly, training those single-modality encoders from scratch will results in slower convergence and need more computation resource which is not economic and not affordable for all researchers. By contrast, utilizing pretrained models as single-modality encoders can serve as a good start point and make model focus more on cross-modality correlations learning, and this paradigm has been widely adopted by most currently main-stream cross-modality pretraining works (such as BLIP, BLIP-2, i-Code, VALOR, etc.).
>
> **weakness 3**
> Thanks for pointing it, we conducted additional ablation experiments  with regards to the contribution of different losses. All models are trained on VAST-27M with "V+A+S" modalities used in both pretraining and finetuning. Due to time limitation we only train half of iterations as those experiments in Table 7 of main paper. The results are shown in the following table from which we can find that all losses (Eq1-Eq4) are important for VAST's superior performance.  The effectiveness of Eq5 has been demonstrated in the Table 7 of main paper.
> | Pretraining Loss | MSR-ret (R@1)| MSR-cap (CIDEr) | MSR-qa (Acc)|
> |---|---|---|---|
> | - | 37.6|  60.3| 44.6
> | OM-VCC  | 43.9 | 61.3 | 45.0 |
> | OM-VCC + OM-VCM | 46.5 | 63.2 | 45.6 |
> | **OM-VCC + OM-VCM + OM-VCG (default settings)** | **47.8** | **68.1** | **46.8** |
>
> **weakness 4**
>
> Due to that web-crawled datasets have limitations in both caption accuracy and diversity, generating training data can supply more high-quality cross-modality samples for training by combining and transfering model’s learned representations into training corpus. This routine has been widely adopted in today’s models such as BLIP, BLIP-2, CLIP-VIP, and MLLMs such as LLaVa and miniGPT4. Combining multiple datasets is a also widely-used way to let model see different data distribution and present overfitting to single corpus.
>
> **question 1**
>
> Considering subtitles (ASR transcriptions in our work) takes also the format of text like captions, we use one single text encoder for both caption and subtitle encoding (share weights for two modalities). An alternative way is to use two separate encoders. From following experiments results we can find using shared one encoder has better performance on five benchmarks and also less total parameters.
>
> | Text and Subtitle encoders| MSR-ret(R@1) | MSR-cap (C) | MSR-qa(Acc) | YouCook2-ret(R@1) | YouCook2-cap(C) |
> |---|---|---|---|---|---|
> | **Share(default settings)** | **47.8** | **68.1** | **46.8** | **41.2**| **179.0** |
> | Seperate | 45.6| 67.8 | 46.3 | 37.7 | 175.0 |
>
> **question 2**
> + Firstly, the V,A,T,S represents for vision, audio, text and subtitle, following the abbreviations of Table 1. Thanks for point it and we will explain this in next version.
> + Secondly, Eq1-Eq4 singly model the relationship between ‘VAS’ and ‘T’, however, as section 4.3 states, in practical scenario  there are many circumstances that only some of modalities but not all  modalities are available. Considering this, we additionally build the relations between V-T, A-T, VA-T and VS-T to keep training-testing consistency and let model performs well at different modality inputs (Eq 5).
>
> **question 3**
> Contrastive loss have already been used in Eq1 (OM-VCC) to pull the global representations of matched samples together. Eq2 is the match loss which infers if a pair of video and caption matched to each other or not. Compared to VCC, VCM encourages more fine-grained interactions between caption tokens, vision or audio patches. Both losses are widely-adopted in current cross-modality researches (usually named ITC and ITM). Above experiments in Weakness 3  also demonstrates the necessity of OM-VCM.
>
> **question 4**
> We conduct following experiments with differnet frames utilized in pretraining. Using more frames can imporve performances a liitle bit. Considering the balance of performance and training cost, we choose to sample only 1 frame during pretraining.
> | Sampled frames | MSR-ret(R@1) | Msr-cap(CIDEr)  | Msr-qa(Acc) |
> |---|---|---|---|
> | 1 (default setting) | 47.8 | 68.1 | 46.8 |
> | 2 | 48.0 | 68.4 | 46.9 |
> | 4 | 48.0 | 68.6 | 47.1 |
>
> **question 5**
> Indeed, in the training process loss ratio of VCM and VCC have already been equal (1:1). And at test process we first compute similarities according to VCC which is fast and then use VCM to rerank top-50 candidates computed by VCC which can further improve the retrieval accuracy, this process is commonly used in latest vision-language pretraining methods such as ALBEF, BLIP, BLIP-2.

---

> > ### Comment · Reviewer_zFyv · 2023-08-17
> >
> > After carefully reading authors rebuttal, I maintain my original rating.

---

> > > ### Comment · Area_Chair_upaY · 2023-08-18
> > >
> > > Dear Reviewer zFyv,
> > >
> > > Thanks for your service as a reviewer and your response to the authors' rebuttal. However, the response to the author rebuttal does not describe how the authors did not resolve your concerns. If there are some elements that you think the authors should have done to resolve your concerns, please tell them as soon as possible. The authors probably can provide more information to rebut further.
> > >
> > > Best,
> > > AC

---

> > > > ### Author Response · Authors · 2023-08-19
> > > > **A complaint to Reviewer zFyv**
> > > >
> > > > Dear AC,
> > > > Thanks for your attention to our paper. Here we would like to make a complaint against **reviewer zFyv** for his/her extreme irresponsible review. We argue his/her professionalism from the following three perspectives.
> > > >
> > > > （1）First, **limited paper reading capability**. He/her reviewed in weakness1 that “paper is not easy to follow”. We think this review is not appropriate due to that **two other reviewers have explicitly reviewed that our paper is easy to follow**. For example, reviewer 1sdz has reviewed that “**The paper is clearly written and easy to read**” and reviewer hHHx has reviewed that “**the paper is well-written with clear explanations**.” So we think probably reviewer zFyv himself/herself dose not have enough paper reading capability and patience.
> > > >
> > > >
> > > > （2）Second, **absurd judgement about “incremental”**. He/Her reviewed in weakness2 and weakness4 that ”Paper is highly incremental”, for the reason that “model have been heavily built on top of numerous existing pertained models” or ”VAST stands on the shoulders of numerous pertained models developed by others”.
> > > > + The whole VAST model or trained vision/audio captioners only takes **three** current models (BERT for text, CLIP for vision and BEATs for audio) for single-modality encoder initialization, which is exaggerated by him/her with words **“heavily”** and **“numerous”**. In addition, those three models we take are very common and used by lots of cross-modality researchers. As stated in our rebuttal to him/her for weakness2, cross-modality pretraining which is the theme of our paper concentrate more on multi-modal connections training instead of single-modality representation training. Using those weights as initialization can be a good start point letting model focus on cross-modality learning, which is also high-efficiency especially for researchers with limited computation source.
> > > > + According to his/her standard to infer if a work is “incremental”, he/her can **reject any paper** even without reading rather than those 1) who happenedly researches image classification, action recognition, audio recognition. 2) who has enough computation source to train all things from scratch (such as CLIP, CoCa). So the review process will be extreme easy that we get a paper, and we directly look into the implement details, and if we successfully find that current work does not belong to above two classes, we just tag it as “incremental” and give a reject score.
> > > > + The contribution of VAST modal and VAST-27M dataset has been illustrated in main paper as well as in [common question 1] in our global rebuttal, and also **recognized by all other reviewers**. For example, reviewer v49r reviewed that ”The new VAST-27M dataset could be very valuable for researchers working on related topics. The experimental results also demonstrate the advantages of the trained model“. Reviewer hHHX reviewed that ”I recognize the contributions of the paper, especially the dataset”. Reviewer “1sdZ” reviewed that "The idea of combining text, video, and audio in one caption is interesting and novel. I believe this will contribute to the multimodal learning community.“  In addition, all reviewer think that the performances of VAST on series of benchmarks are strong.
> > > >
> > > > (3) Third, **unfamiliar with cross-modality pretraining’s common objectives**. Two questions (Q2 and Q5) proposed by him/her concentrate on contrastive and matching loss, which are two extreme common losses widely used in cross-modality pretraining field for relatively a long time. VAST uses them to strengthen both coarse-grained and fine-grained alignment. In Q2, he asks the necessity of matching loss and in q5 he asks what if  ratio of two losses are 1:1, however, the ratio of two losses are just 1:1 as shown in Eq4 of the main paper. Those questions reflect that he/her **has not read paper carefully** and is also **not familiar with the basic knowledge of cross-modality pretraining**.
> > > >
> > > > (4) Fourth, **invaluable feedback**. We sincerely thanks for NIPS that support a discussion period that let authors and reviewers communicate to each other. The authors have **carefully answered his all questions and shown many additional experiment results in the rebuttal stages**. However, as you commented, he/she commented with simply “maintain my original rating” without any reason or further questions.
> > > >
> > > > In a word, we think it is not appropriate to give a 3 score reject rating to our paper with such subjective reasons (individually fell not easy to read / individually feel incremental), and we think reviewer    zFyv  is not professional enough to serve as a reviewer. We sincerely hope AC could judge this paper and reviewer zFyv ’s review fairly, thanks very much!
> > > >
> > > > Best,
> > > > Author

---

### Official Review · Reviewer_hHHX · 2023-07-06

**Soundness:** 3 good
**Presentation:** 3 good
**Contribution:** 3 good
**Rating:** 6
**Confidence:** 5

**Summary:**

This paper introduces a large-scale dataset called VAST-27M, which contains audios, videos, subtitles, and generated captions. The authors employ an LLM to combine all forms of textual information to create omni-modality captions. They train an omni-modality video-text foundational model on this dataset. Extensive experiments conducted on various downstream tasks demonstrate the effectiveness of both their dataset and model.

**Strengths:**

- They provide a valuable dataset, and the experiments demonstrate the superiority of this dataset.
- The paper is well-written with clear explanations.
- The experiments are thorough, including the validation of the methodology, the effectiveness of the dataset, and comprehensive comparisons with the state-of-the-art (SOTA) approaches.
- The model achieves state-of-the-art (SOTA) performance on 22 downstream tasks.

**Weaknesses:**

- The authors utilized the Vicuna model instead of ChatGPT, which might have an impact on the quality of the data. Could you provide a comparison of the quality of data generated by ChatGPT and Vicuna?
- Why wasn't there consideration for multimodal LLMs like LLaVa or MiniGPT-4, and instead, but using a separate trained captioning model?
- The methodology lacks technical contributions and doesn't differ significantly from previous works in terms of model architecture and training. When dealing with multiple forms of text simultaneously, the current models simply aggregate multiple losses, which may not fully leverage the available information.


**Questions:**

I'd like to inquire about the authors' plans for sharing the data and code, including the captioning model, VAST model, and the dataset.

**Limitations:**

Yes.

---

> ### Author Rebuttal · Authors · 2023-08-08
>
> **weakness 1**
> + Our work needs to generate  27M omni-modality captions, and locally deployed open-resource large language modal (such as Vicuna) can fulfill parallel generation which is efficient. However this is not feasible for ChatGPT due to its API is relatively inefficient and requies additional costs, so we choose to use the former for caption generation.
> + With regards to caption quality comparison between Vicuna and ChatGPT, we visualize two examples in the following table. The same video caption,  audio caption,  subtitle and instruction are given to both models and we sample omni-modality captions twice. From the examples shown in the following table, we can find that both models can generate relatively high-quality captions and can comprehensively reflect infomation of all modalities. Captions generated by ChatGPT tends to be longer and cover most clues given in the prompts, and can also imagine scences like "wind rustled through the trees" while 'trees' does not appear in the prompts. By contrast, Vicuna tends to generate short but accurate captions, while some details like "splashing" and "mailbox" may be missed.
> |Modality|Caption|
> |---|---|
> | Audio| Wind, birds, women speaking, splashing, and human sounds are heard. |
> | Video| Woman picked up a phone and walked around the garden. |
> | Subtitle | I can't take delivery now. I'm on vacation, why don't you put it next to the mailbox at the door of my house. |
> | Omni(by Vicuna-13b)  | A woman walks with a phone in a garden while birds chirp, wind blows, and a human speaks, with the sound of splashing in the background, and the delivery cannot be taken now as the person is on vacation, so they suggest leaving it near the door of their house. |
> |  | A woman walks with a phone in a garden while wind, birds, and human sounds are heard, and a person is unable to receive a delivery due to being on vacation. |
> |  Omni(by Chat-GPT) | The woman, while on vacation, picked up her phone and walked around the garden, surrounded by the sounds of wind, birds, splashing, and various human sounds. In a speech subtitle, she mentioned being unable to take delivery and suggested leaving it next to the mailbox at the door of her house. |
> |  | The woman, who was on vacation and unable to take delivery, walked around the garden, speaking into her phone, while wind rustled through the trees, birds chirped in the background, and various splashing and human sounds filled the air. She suggested leaving the package next to the mailbox at the door of her house. |
> |**Modality**|**Caption**|
> | Audio | A turkey is heard, followed by coughing, wind, bird chirping, and animal sounds. |
> | Video | Teenage boys walking in a zoo togehter on a narrow pathway surrounded by trees. |
> | Subtitle| Where do we go now, shall we see elephants first? But I heard that the gorilla exhibition is going to start soon, why don\'t we go see that first. whatever, we have plenty of time today. |
> |Omni(by Vicuna-13b) | Two teenage boys walking in a zoo surrounded by trees and trees, with a turkey and various animal sounds in the background, discussing their next destination. |
> |  | In a zoo, teenage boys walk together on a narrow path surrounded by trees, while a turkey is heard, followed by coughing, wind, bird chirping, and animal sounds, and they discuss where to go next, considering seeing elephants or the gorilla exhibition. |
> | Omni(by Chat-GPT) | As the teenage boys walked together on a narrow pathway surrounded by trees at the zoo, a turkey's call echoed in the background, followed by the sound of coughing, gentle wind, bird chirping, and a symphony of animal sounds. One of the boys remarked, "Where do we go now? Shall we see the elephants first? But I heard that the gorilla exhibition is going to start soon. Why don't we go see that first? Whatever we decide, we have plenty of time today." |
> |  | As the teenage boys strolled through the zoo surrounded by trees, the air was filled with the sound of a turkey, accompanied by the occasional cough, gentle wind rustling the leaves, the cheerful chirping of birds, and a variety of other animal sounds. Amidst their conversation, one of the boys pondered their next move, suggesting they start by visiting the elephants. However, another boy excitedly mentioned that the gorilla exhibition was about to begin and proposed going there first. Regardless of their decision, they reassured each other that they had plenty of time to explore the zoo today. |
>
> **weakness 2**
> + Firstly, this work(VAST) has been done concurrently with LLava and MiniGPT-4. In addition we think that even though MLLMs can generate more detailed descriptions, but their caption accuracy may be not promised (such as the common hallucinations which is a general problems in those models), and inaccurate captions may have bad effects.
> + Secondly, the inference speed of those models are limited compared to trained captioners due to their extremely large hyperparameters.
> + Thirdly, there seems not be available audioLLMs for generating audio captions.
> + Fourthly, LLava and MiniGPT-4 use image-text data for pretraining which may not be appropriate for captioning videos that concentrate on both static appearances and dynamic motion informations.
>
> Considering above reasons, we choose to train captioners on mixed datasets (both image and video data) to promise its caption quality.
> **weakness 3**
> The technical contributions of VAST have been illustrated in the global rebuttal [Common Question 1].
> **question**
> We promise that all of VAST-27M data (vision captions, audio captions and omni-modality captions), model checkpoints, training and testing codes of VAST will be public released to facilitate research of multimodal learning communities.

---

> > ### Comment · Reviewer_hHHX · 2023-08-18
> >
> > The author's reply solved my concerns and I recognize the contributions of the paper, especially the dataset.

---

> > > ### Author Response · Authors · 2023-08-21
> > >
> > > Dear reviewer,
> > > Thanks for the reconition of our work, and also the effort and time you have paid  serving as a reviewer!
> > >
> > > Author

---

### Official Review · Reviewer_v49r · 2023-07-12

**Soundness:** 3 good
**Presentation:** 2 fair
**Contribution:** 3 good
**Rating:** 5
**Confidence:** 4

**Summary:**

The paper proposed to build a new multi-modal foundation model that includes vision, audio, subtitle and text modalities. The designed architecture of the model is very similar to previous works, such as BLIP. The paper also made another contribution, a new multi-modal video dataset VAST-27M. VAST-27M reuse videos from another dataset HD_VILA_100M. The paper uses off-the-shelf video captioning models and audio captioning models to generate text descriptions for the input video. Large language models are then applied to summarize the text descriptions together with the original subtitles of the video using customized prompts. The paper presents experimental results on 22 downstream datasets to demonstrate the effectiveness of the proposed approach.

**Strengths:**

High-quality pre-training datasets that are openly accessible to the research community is very important for us to experiment with new research ideas and push the research boundaries. The new VAST-27M dataset could be very valuable for researchers working on related topics. The experimental results also demonstrate the advantages of the trained model.

**Weaknesses:**

The following are more detailed comments and suggestions about the paper.

1, What is the difference between subtitles and text? The paper seems to treat them as different modalities. Also does the audio modality already contain all the information from subtitles?

2, Based on the implementation  details in Section 5.1, the final model is initialized with other pretrained encoders, and pretrained with lots of other datasets as well. It is not clear how much the improvement is from the proposed VAST-27M dataset. It will be better if the paper can show the difference for with or without using the VAST-27M dataset. This can help us better understand the quality of VAST-27M.

3, How to select a subset of 27M videos from HD_VILA_100M? Please elaborate the details why the paper selected a subset from HD_VILA_100M.

4, In Line 60, why does the cross-attention layer only involve the text modality? Do we have cross-attention between video and audio?

5, The paper seems to use BLIP for vision captioning. Why not use BLIP-2, which can generate better captions?

6, The overall design of the learning objectives is very similar to BLIP, which also uses three losses, i.e., Image-Text Contrastive Loss (ITC), Image-Text Matching Loss (ITM), and Language Modeling Loss (LM). The novelty of the proposed learning framework seems to be very limited.

7, All the captions from VAST-27M are generated by models. Does the paper look into the bias of the used models and their generated captions? Will they limit the quality of the dataset?

8, A lot of details in the paper are similar to reference [7]. It is unclear what is the real contribution of the current submission. Many sections are very similar to [7]. I recommend clarifying the difference with [7], e.g., what are the contributions from the current submission, what has been done in [7].

**Questions:**

See above.

**Limitations:**

See above.

---

> ### Author Rebuttal · Authors · 2023-08-08
>
> **weakness 1**
> + Subtitle is usually achieved by ASR or OCR techniques and can be a supplementary modality for video understanding, while captions are usually objective descriptions about objects existed or events happened in images or videos. Even though they are both represented as text, they have different data distribution and information density. In this work, the word "T"/"text" in VAST refers to "caption" which is a language output or retrieval query, while "S"/"subtitle" is an input modality. We use text(T) instead of caption(C) in "VAST" simply for getting a abbreviation that sounds better.
> + In this work, audio modality already contains all information from subtitles (ASR texts), but ablation study in Table 7 shows that explicitly taking subtitle as input achieves better results than taking only audio signal as input, and we attribute it to that current audio encoder doesn't possess strong speech reconigiton ablities and we think it's a valuable future research direction to perceive both environment sounds and human speech information from audio signal only.
>
> **weakness 2**
> Thanks for your suggestions. We conduct the following experiments to demonstrate the effectiness of VAST-27M. VAST-B is trained on same datasets with VAST but vision backbone replaced with CLIP-VIT-B/16 for fast training. Both two models are trained for same iteration and batch size. The results in following table shows that VAST-27M contributes a lot to VAST's high performance, without which the performances drops drastically on seven benchmarks.
> | Model | MSR-ret | MSR-cap | MSR-qa | YouCook2-ret | YouCook2-cap | VALOR32K-ret | VALOR32K-cap |
> |---|---|---|---|---|---|---|---|
> | **VAST-B**  | **56.9/79.8/87.1** | **74.2** | **48.2** | **36.3/57.7/67.3** | **186.7** | **70.0/90.2/94.7** | **54.6** |
> | VAST-B (w/o VAST-27M) |  50.8/76.1/85.5 | 70.7 | 47.0 | 32.6/57.5/67.8 | 130.3 | 62.8/86.5/92.5 | 53.0 |
>
> **weakness 3**
> The specific selection standard has been illiustrated in the global rebuttal [Common Question 2]
> **weakness 4**
> In VAST, text encoder have taken the place of fusion interface for multimodal  understanding and generation via cross attention. Text takes the role of query while single one or some combinations of vision, audio and subtitle  take the role of key and value. We take this design mainly because most tasks we target at (captioning, QA, retrieval, etc.) can be uniformly modeled with language interface. In current version of VAST video or audio has not taken the role of fusion interface and  we think is important to update VAST in future to handle more tasks like language-guide video/audio prediction/generation/edition and then it will be necessary to add cross attention layers in video /audio encoder and also take them as output interfaces.  Thanks for your suggestions.
> **weakness 5**
>  In fact, VAST uses self-trained vision captioner as illustrated in Section 3.1 (line128). We do not use open-source BLIP or BLIP-2 mainly because they are finetuned using image-text data (MSCOCO) only while the capabilities of describing dynamic events or actions are ignored. So we resort to trained a more general captioner that focus on both static or dynamic vision scenarios by finetuning on mixture of image/video captioning datasets, in order to generate more high-quality captions for VAST-27M.
> **weakness 6**
> About technique novelty has been elaborated in the global rebuttal [Common Question 1].  BLIP [6] is a image-language pretraining method aiming at bootstrapping model with trained captioner and filter. The differences between VAST and BLIP can be summarized as follows.
>
> + About data generation. The vision captioner  for VAST-27M is inspired by BLIP, but we make several modifications to fit multi-modality pretraining tasks including 1) training vision captioner with the mixture of image and video caption datasets to increase captioner’s generality. 2) promoting from vision-language field to audio-language filed and trained a  general audio captioner. 3) utilizing LLM to integrate multi-perspective captions.
>
> + About  model architecture. Different from BLIP that contains a image encoder and text encoder, VAST contains additionally an audio encoder which supports audio input. In addition, vision encoder of VAST can support both image and video input, and text encoder can support both caption and subtitle input.
>
> + About  training objectives. Image-text contrastive learning(ITC), matching(ITM), and image-conditional text generation are three most common training loss taken by current image-language pretraining models including BLIP.  However, only relationship of image and text can be constructed through these losses, VAST have successfully modified them to omni-modality losses (OM-VCM, OM-VCC and OM-VCG as illustrated in Section 4.2), and use modality grouping loss illustrated in Section 4.3 to handle different modality combinations, which can not be realized with BLIP’s architecture and training objectives.
>
> **weakness 7**
> The dataset bias has been illiustrated in the global rebuttal [Common Question 3]
> **weakness 8**
> VALOR proposed a one-million level video dataset that connects text with both video and audio, and  a vision-audio-text model. VAST has explicit differences and strengths compared with VALOR, which can be summarized as follows:
>
> + Firstly, VALOR-1M contains limited semantic concepts and cannot be friendly scaled up due to it's manual annotaion . By contrast, VAST27M is automatically generated that  can be scaled up easily, and contains much more semantic concepts, which can be quantitatively reflected by experiments shown in Table 4,5,6 in the paper: models pretrained on VAST-27M outperform models trained on VALOR-1M on V-T, A-T and AV-T tasks.
>
> + Secondly, as illustrated in Common Question 1, VALOR stands for a series of works that models both vision and audio while VAST for the first time unifies all those modalities in one framework.

---

> > ### Comment · Reviewer_v49r · 2023-08-21
> >
> > Thank the authors for answering my questions during rebuttal. Some of my questions have been clarified by the authors. However, I am still skeptical about  weakness 4 and  weakness 8. In weakness 4, ignoring the audio and video in cross attention is not well justified, as the paper claims to be Omni-Modality Foundation Models. For weakness 8, the contributions of the current submission are limited if we consider reference [7] as pre-exist works.

---

> > > ### Author Response · Authors · 2023-08-22
> > >
> > > Dear reviewer,
> > >
> > > Thanks for your reply! Due to that in this work we mainly focus on cross-modality semantic understanding and generation tasks which take language as output interface such as retrieval, captioning and question answering, regarding model architecture design we simply take language as the query while vision, audio and subtitle as key for cross-attenion mechnism in text encoder. In the future work, we will improve the foudation model to additionally support visual/audio/audiovisual grounding tasks, like moment retrieval or text-conditioned highligt detection. To support those tasks, a cross attention inferface for audio or vision may be needed, or we can simply follow DETR to initailize a series of learnable tokens that concatenate with text tokens, and predict locations at the output of those learnable tokens. Thanks for your valuable advice!
> > >
> > > In addition, our work has big differnece with VALOR. First, with regards to the dataset, VAST-27M is a **auto-generated** dataset which is **27 times larger than VALOR-1M**, which is a manual annotated dataset. With regards to the foundation model, as illustrated in the global rebuttal **[common question 1]**, VALOR can be categoried in the first class which models vision, language and **audio** jointly (e.g. AVLNet, i-Code, VALOR , ECLIPSE). There is another series of works that model vision,language and **subtitle** jointly (UniVL, MELTR). VAST for the first time **unifies both two series of works and support all of tasks that each of them can support**. In addition, VAST achieves 22 new state-of-the-arts results, and **surpasses VALOR on all tasks that VALOR has evaluated on**, with huge margins.
> > >
> > > Best,
> > >
> > > Author

---

> ### Author Response · Authors · 2023-08-21
>
> Dear reviewer,
> there is not so much time left for the discussion stage, if you still have questions about our work, please let us know and we will reply as soon as possible, thanks for your effort and time!
>
>
> Author

---

### Official Review · Reviewer_1sdZ · 2023-07-14

**Soundness:** 3 good
**Presentation:** 3 good
**Contribution:** 3 good
**Rating:** 5
**Confidence:** 5

**Summary:**

This paper proposes a large-scale omni-modality video caption dataset based on an existing video-text dataset, namely VAST-27M. Specifically, the authors utilized the VL models to generate captions and LLM to generate omni-captions for videos. Then a vision-audio-subtitle-text model is trained for a range of video tasks and establishes SOTA in several benchmarks.

**Strengths:**

The paper is clearly written and easy to read.
The idea of combining text, video, and audio in one caption is interesting and novel. I believe this will contribute to the multimodal learning community.
The performances over a wide range of video-language-audio tasks are strong.


**Weaknesses:**

1. Despite the dataset contribution, the technical contribution of the pre-training model is somewhat weak and lacks novelty.
2. I am confused by some details in the approach. (1) In Lines 185-186, why do you need to feed the caption tokens again instead of using the textual embeddings from the text encoder above? (2) In OM-VCG, do you use the masking strategy like BERT or generative training like GPT? Based on my understanding, you are adopting the former one, for which I think the name of “video caption generation loss” is improper, masking language modeling (MLM) may be better.
3. Although the experimental results are very strong, I am unsure where the performance gain comes from. Several points that might be attributed to this are not discussed. (1) Combination of different datasets. The model combines several different datasets, including VAST-27M while the dataset scale is not compared with SOTA methods. (2) Strong backbones. As indicated in Line 213, EVAClip-ViT-G is adopted as a visual backbone which is different from SOTA models, which makes the comparison in Table 3 unfair. (3) More modalities. Although Table 7 provides an extensive comparison of using different modalities, there are no results that could be directly compared in Table 3 for a fair comparison.
4. Some details are missing. (1) How do you select the 27M video clips from 100M clips in HD-VILA-100M? What is the selection standard? (2) Do you use audio in the HD-VILA-100M dataset for audio captioning and as input for the audio encoder? It is not clearly explained in the paper. (3) How do you sample frames in videos and how do you deal with video embedding from frame features in ViT (i.e., how to get f_v)? By Meanpooling?
5. Lack of discussion on related works. As an extension of HD-VILA, CLIP-ViP used an off-the-shelf image captioner to generate captions for the HD-VILA-100M dataset. The authors should also compare that one and your advantage over CLIP-ViP. It seems that only audio information is not used in the CLIP-ViP model.
6. It would be helpful to show some examples in the dataset for illustration.
7. Some typos: Row 1 in Table 5.


**Questions:**

Please see the weakness part.

**Limitations:**

The authors have addressed the scale limitation of the current dataset. It would be great if they could share more insights into the potential limitations of using their dataset and model, such as biases it could contain.

---

> ### Author Rebuttal · Authors · 2023-08-08
>
> **weakness 1**
> About the technical contributions of VAST model have been explained in our global rebuttal **[Common Question 1]**.
>
> **weakness 2**
> (1) Using origin captions in OM-VCM, OM-VCG and OM-VCC can keep consistence with BERT whose parameters are inherit for text encoder initialization. In addition, we have also tried use text embeddings of OM-VCC as input for text encoder used in OM-VCM and OM-VCG, but found performance slightly reduced.
>
> (2) We assume that common generative modeling method can have two forms: generative modeling like GPT style (LM, input current word, output next word), and causal masked language modeling (CMLM, mask current token and predict it, but with only previous tokens as conditions). As stated in Section 4.2 (line196), VAST takes the latter one (CMLM) for the reason that in the early development stage we found that BERT+CMLM performs slightly better at most downstream benchmarks than GPT+LM. We name it OM-VCG from the target perspective that is to generate captions, and it can also be renamed as OM-CMLM.
>
> **weakness 3**
> (1) For the space limitation reason, we put more detailed comparison results including the dataset scale comparison among different SoTA methods in Appendix.
> + From the results in Table 4 and Table 5 (Appendix), we can find that VAST (442M data) outperforms CLIP-VIP(500M),  Florence(900M) on text-to-video retrieval benchmarks with huge margins.
> + From the results in Table 6 (Appendix), we can find that VAST (442M) outperforms VideoCoCa(4.8B), GIT2(12.9B), Flamingo(2.3B) on video QA benchmarks with huge margins.
> + From the results in Table 7 (Appendix), we can find that VAST (442M) outperforms GIT2(12.9B), GIT(1.7B), MaMMUT(2B) on video caption benchmarks with huge margins.
>
> (2)  To get the best performance and demonstrate our methods function at the most advanced vision models, we use a strong vision backbone (EVACLIP-Giant) in the main paper, which is not consistent with other methods. To make fair comparison, we additionally trained a VAST-B model which use CLIP-VIT-B/16 as backbone and compare it to  methods with the same backbone used. The results  are shown in the following table which demonstrated that VAST can outperform both only-vision and multi-modal video-language pretraining models on three representative benchmarks.
>
> | Model | Vision Backbone | MSRVTT-Ret(R@1/R@5/R@10) | MSRVTT-Cap(CIDEr) | MSRVTT-QA(Acc) |
> |---|---|---|---|---|
> | GIT2-B | CLIP-VIT-B/16 | - | 57.8 | 41.0 |
> | mPLUG2-B | CLIP-VIT-B/16| 48.3/75.0/83.2  | 72.4 | 46.3 |
> | UMT-B | CLIP-VIT-B/16| 51.0/76.5/84.2 | - | 44.9 |
> | VALOR-B | CLIP-VIT-B/16| 43.0/72.2/82.1 | 66.6 | 46.7 |
> | **VAST-B (Ours)**  | CLIP-VIT-B/16| **56.9/79.8/87.1** | **74.2** | **48.2** |
>
> (3) In the Table 4,5,6,7 of Appendix, a groups of methods utilizing multiple modalities beyond vision and text (audio, subtitle) are marked by gray background colors and we mainly compare VAST with those groups of methods for fair comparison. From the results we can find that VAST outperforms all not only single-modality but also multi-modality methods.
>
> **weakness 4**
> (1) The specific selection standard has been illiustrated in the global rebuttal **[Common Question 2]**
>
> (2) Audios in HD-VILA-100M are not used in audio captioner training cause that the subtitle is ASR transcriptions rather than objective captions. Instead we use the combination of VALOR-1M and WavCaps datasets, as illustrated in Section 3.1 (line136). In addition, audios in VAST27M (sourced from HD-VILA-100M) is used as input for VAST model pretraining.
>
> (3) As illustrated in section 5.1 (line219), during  pretraining we sample only one frame for efficiency and during finetuning, we sample different frames (8~32 frames) for different benchmarks as shown in Table 3 in appendix. With regards to f_v, we do not conduct mean pooling but simply concatenate patch features of all frames together to reserve most information for multimodal encoding or decoding. The computation cost is affordable due to cross-attention mechanism (linear complexity for condition features) used in multimodal fusion instead of merge attention (quadratic complexity). We are sorry about not clarifying f_v, and will modify it in next version of paper, thanks for pointing it.
>
> **weakness 5**
> + CLIP-VIP  proposes to re-generate captions for HD_VILA_100M dataset through open-source caption model OFA, and use both raw subtitle and generated captions for video-language pretraining. Compared with CLIP-VIP, VAST has explicit differences and strengths which can be summarized as follows.
> + Firstly, OFA  which is the captioner CLIP-VIP used is a image-text model which has not seen video caption data during training, while our video captioner is trained on mixture image-text and video-text corpus that can generate more motion-related video captions. In addition, their captions reflects visual contents only while captions in VAST-27M contains both vision, audio, subtitle descriptions and omni-modality caption, which is apparently more appropriate for multi-modality pretraining, given its importance as discussed in Common Question 1.
>
> + Secondly, CLIP-VIP has tried use together generated visual captions and raw subtitles and models both "V-T" and "V-S" relationships while VAST models VS-T correlations. This is because we think at most practical scenario we would like to put subtitle on the video input to help video content understanding, instead of predicting them which can be independently addressed by mature ASR algorithms. In addition, CLIP-VIP does cross-modality tasks (caption, QA, retrieval, etc.) only based on vision signal (1 signal), while VAST can utilize vision, audio and subtitle (3 signals).
>
> **weakness 6**
> Considering the limited space in main paper and many contents need to be clarifed, we put more examples in the Appendix (Page 9 B.3).
>
> **weakness 7**
> Thanks for pointing it, we will change "PT Data" to "Model"  in the next version.

---

> > ### Comment · Reviewer_1sdZ · 2023-08-17
> > **Response to rebuttal**
> >
> > Thanks for the authors' comments. I will keep my score.

---

> > > ### Author Response · Authors · 2023-08-21
> > >
> > > Dear reviewer,
> > > Thanks for the reconition of our work, and also the effort and time you have paid  serving as a reviewer!
> > >
> > > Author

---

### Author Rebuttal · Authors · 2023-08-08

We sincerely thanks all reviewers for your recognition of our work VAST (dataset and foundation model), and very insightful reviews. Your reviews and suggestions are very important for us to improve VAST model and paper. We will add necessary detail descriptions and model comparisons in the next version of paper. With regards to rebuttal, we will firstly make an overview reply for some common questions proposed by at least two reviewers, and then make reply for others individually.

**[Common Question 1] About VAST model’s technique novelty and contribution**

VAST is the first onmi-modality pretraining model which aims at  building cross-modality connections between vision, audio, subtitle and text, and supporting both single vision-language, multi-modal video-language  and audio-language tasks. We think VAST’s innovations can be summarized as follows:

+ Under the scenarios that current multi-modal video-language models are somewhat specialist either additionally supporting audio modality beyond vision and language (i-Code [1], VALOR [2], etc) which can tackle audio-related tasks, or supporting subtitle modality (UniVL [3], MELTR [4],etc) that enhance the understanding of human speech, recipes or news, VAST is the **first** generalist foundation model that can support all vision (both image and video), audio, subtitle and text as input.

+ VAST have **firstly** soundly proven two things through extensive large-scale pretraining-finetuning ablation experiments. Firstly, utilizing all modalities (vision, audio, subtitle) can get consistent improvement over a very broad range of different types of modality-oriented downstream tasks (Table 7 line-j vs line-e). Secondly, the multi-modality enhancement functions more apparently with the help of large-scale pretraining (Table 7 line-j&line-e vs line-d&line-a), which demonstrated the meaning and necessities for omni-modality pretraining research.

+ VAST has achieved absolutely dominated SoTA performances on 22 cross-modality benchmarks (image-language, video-language and audio-language benchmarks) which is kept till now. We think this to some extent shows that compared to further promote VLP architecture or training objectives, bringing in more modalities is a much bigger cake, but how to enjoy this cake is relatively less researched. We hope that VAST can attracts more attention over the process of "steping further from conventional vision-language pretraining towards omni-modality pretraining", and we promise that all datasets, codes and model checkpoints will be publicly released to help communities for reproduction and further research.


**[Common Question 2]  About data selection rule of VAST-27M from HD_VILA_100M.**

+ Videos in HD_VILA_100M [5] dataset covers a broad distribution of categories including music, gaming, sports, film, animals, education, entertainment, etc. After original processing in [5], each pair in the HD-VILA-100M consists of a video clip about 13.4 seconds on average and a sentence with 32.5 words on average. We choose to use a subset of HD_VILA_100M instead of using all of them mainly due to following three reasons: 1) the comprehensive consideration about the time and money cost of downloading, storage and computation. 2) those 100M clips originate from 3.3M long videos, which means that many clips are from the same video and share similar scenes. 3) many of them cannot be downloaded from YouTube.

+ In fact, we select the videos from HD_VILA_100M mainly according to following 4 rules: 1) we filtered out clips whose length are too short (less than 5 seconds) or too long (more than 30 seconds). 2) we filtered clips whose one or some modalities of vision, audio and subtitle are missing. 3) similar numbers of clips are sampled from every video in those 3.3M ones. 4) for each long video, the sampled clips are as disperse as possible to each other to ensure diversity and reduce redundancy. To the end, we got 27M video clips with an average duration of 10.0 seconds. The average lengths of vision, audio, and omni-modality captions in VAST-27M are 12.5, 7.2, and 32.4, respectively.


**[Common Question 3]  About potential data bias of VAST-27M.**

+ Due to the reason that the data collection process of VAST-27M have used a large language model (Vicuna-13b), and the training process of video and audio captioners have utilized some open-sourced cross-modality corpus such as CC4M, CC12M, LAION, VALOR-1M and WavCaps, so VAST-27M and VAST model may suffer the same bias of those dataset and bias of Vicuna-13b.

[1] Yang et al. i-code: An integrative and composable multimodal learning framework.
[2] Chen et al. Valor: Vision-audio-language omni-perception pretraining model and dataset.
[3] Luo et al. Univl: A unified video and language pre-training model for multimodal understanding and generation.
[4] Ko et al. Meltr: Meta loss transformer for learning to fine-tune video foundation models.
[5] Xue et al.Advancing high-resolution video-language representation with large-scale video transcriptions.

---

### Decision · Program_Chairs · 2023-09-21

**Decision:**

Accept (poster)

**Comment:**

The contributions of this paper are two-fold: 1) proposing a new dataset creation pipeline and a large-scale multimodal video dataset, VAST-27M as a result, and 2) building an omni-modality foundation model that achieves new SOTA performances on 22 benchmarks.

The paper received positive scores from three reviewers (6, 5, 5) and a negative score from the other (3). The three positive reviewers recognize their technical contributions, especially on the dataset creation side. In my understanding, this is the first dataset consisting of all three modalities (vision, audio, and subtitle) and additional captions. Having this additional caption should be properly appreciated as it is critical to have another text for each example to use subtitles as a part of the multimodal inputs.

Although it is concerned that the method is highly incremental and relies too much on existing models both for building a new model and for the dataset creation, it is a common practice to use these existing unimodal pretrained models as a starting point to build a larger multimodal model. Also, using highly-performing captioning models to build a new benchmark if needed for omni-modality training can still be novel.

Overall, the AC thinks that the merits outweigh the flaws of the paper, and would like to recommend acceptance of the paper.